# NEXT-TOBE: PROBABILISTIC NEXT TOKEN-BAG EXPLOITATION FOR ACTIVATING ANTICIPATORY CAPACITY IN LLMS

**Yihe Liu**[1][*]**, Huibin Wang**[1][*]**, Xianming Hu**[1][*]**, Pinyi Zhang**[1]**, Jiahao Xiong**[1]**, Chenglin Wang**[1]**,
Nuoyi Chen**[2]**, Hongbo Zhao**[1]**, Jie Zhang**[2]**, Kai Zhang**[1][†]
[1] School of Computer Science and Technology, East China Normal University
[2] Institute of Science and Technology for Brain-Inspired Intelligence, Fudan University

## ABSTRACT

Auto-regressive large language models (LLMs) exhibit a non-trivial capacity to "anticipate" long-range future tokens despite being trained to predict only one token at a time. Nevertheless, how to systematically profile, enhance and leverage such capacity to improve LLM reasoning performance remains unclear. In this paper, we propose **Next Token-Bag Exploitation (Next-ToBE)** to tackle this challenge. Next-ToBE quantifies LLM's anticipatory capacity by measuring how well tokens within a predefined future window are pre-captured by the model's current softmax probabilities. This capacity is strongly correlated with LLM generative quality but often suppressed by the rigid one-hot objective in next-token prediction. To address this, we replace the one-hot target vector in next-token prediction with a soft target distribution spanning additional future tokens. Specifically, the immediate next token retains the highest importance, while more distant "look-ahead tokens" are also included to enrich supervision, with their importance dynamically determined by temporal and semantic relevance patterns to inject forward-looking pressure. Besides, the fitting process emphasizes the model's intrinsic anticipatory tendency, thus preserving the confidence and fidelity of the pre-trained model to improve training stability. Overall, Next-ToBE not only effectively activates LLM anticipatory capacity through fine-tuning, yielding notable gains in reasoning performance with higher memory and computational efficiency against the MTP baselines, but also shows great potential in pretraining setting by successfully cultivating this capacity from scratch. These highlight its value as an effective strategy to extend the prediction horizon of LLMs, enabling them to see further, and reason better.

## 1 INTRODUCTION

Large language models (LLMs) built on the auto-regressive next-token prediction (NTP) (Williams & Zipser, 1989) have achieved remarkable success across diverse language modeling tasks (Yang et al., 2025; Abdin et al., 2024). Although NTP is restricted to generating only one token at a time, recent studies suggest that LLMs trained under this paradigm also exhibit anticipatory capacity, implicitly forecasting tokens beyond the immediate next one. For example, Pal et al. (2023) show that intermediate-layer hidden states can predict future tokens; Dong et al. (2025) report that LLMs can encode global response properties even before producing the first token; Wu et al. (2024) further show that this is attributed to gradients from future-token losses shaping earlier predictions.

The anticipatory capabilities of LLMs mirror the human cognitive process of internally rehearsing before speaking (Barthel et al., 2016), i.e., "think first, generate later", which can be critical for tasks requiring multi-step planning and long-range logical consistency, such as mathematical reasoning (Ahn et al., 2024), code generation (Jiang et al., 2024), and human-computer interaction (Li et al., 2023). However, emerging evidence suggests that the next-token prediction (NTP) paradigm may

---

[*]These authors contributed equally to this work.
[†]Corresponding author.

inadvertently suppress this capacity. Deng et al. (2020) showed that the one-hot fitting objective in NTP inherently biases the model toward the marginal distribution of individual tokens, limiting its ability to capture long-range semantic and temporal dependencies. Such emphasis on step-level correctness could promote short-range pattern matching and undermine multi-step planning ability (Bachmann & Nagarajan, 2024). Finally, penalizing plausible high-confidence alternatives (e.g., synonyms or paraphrases) deviating from the single ground-truth label reduces the model's semantic flexibility and undermines its ability to maintain coherent internal representations (Li & Lu, 2021).

To address these limitations of NTP, many work have focused on the new paradigm of "multi-token prediction" (MTP). MTP employs multiple parallel auxiliary output heads to predict several future tokens simultaneously (Qi et al., 2020; Gloeckle et al., 2024; Cai et al., 2024). Subsequent enhancements use sequence modeling to tighten long-range dependency (Liu et al., 2024a), improving sequence generation consistency by predicting non-adjacent tokens (Liu et al., 2025) or by masked-generation and knowledge-distillation (Samragh et al., 2025). Meanwhile, efforts within NTP training still continue: Gerontopoulos et al. (2025) incorporate register tokens into input sequences to share memory of future label tokens in advance; and Zuhri et al. (2025) propose an auxiliary loss to predict the rank of upcoming label tokens based on their distance from the current step.

Although these advances offer substantial advantages, some concerns remain. First, higher computational and memory demands could be entailed, like additional parameters (prediction heads), register tokens or auxiliary loss calculation. Second, these approaches still rely on one-hot fitting objective (including MTP), thus inheriting some of its intrinsic limitations like reduced semantic flexibility or over-concentration of predicted probability mass. Third, few studies have systematically profiled the anticipatory behavior in LLMs, making it difficult to clarify its role in shaping model performance, or to effectively enhance it for meaningful gains in real-world applications.

In this paper, we introduce **Probabilistic Next Token-Bag Exploitation (Next-ToBE)**, a simple method to profile, refine, and harness the anticipatory capacity of LLMs to enhance reasoning performance. The main idea is to leverage the LLM's soft-max predictions at each step as an observable proxy for its internal anticipatory signal, and to promote consistency between this implicit planning and actual sequence of future tokens within a predefined window. Concretely, this is achieved by replacing the conventional *one-hot objective* of next-token prediction with a *flexible token-bag distribution* constructed from the window of upcoming tokens. The construction of this distribution follows three principles: (1) The immediate next token is given the highest importance to preserve local coherence and prevent semantic drift. (2) The model's intrinsic anticipatory tendencies is respected to preserve stability and fidelity of the pre-trained model in fine-tuning setting. (3) Distant future tokens are dynamically weighted by their joint semantic-temporal relevance to mitigate noise from loosely related or contextually unstable future tokens.

Overall, Next-ToBE provides rich supervision that activates and refines the internal anticipatory capacity of LLMs. It is simple to implement, requiring only a modification of the target distribution within the standard next-token prediction framework and introducing no additional parameters. Experiments across the Qwen and Llama model families demonstrate that, on challenging reasoning tasks, base models fine-tuned with Next-ToBE achieve consistent and notable performance improvements over strong MTP baselines, particularly in long-horizon and multi-step reasoning scenarios. Similar performance gains are also observed in the pretraining setting.

The contributions of this work are summarized as below:

1. We propose Next-ToBE, a method to profile and activate inherent planning capacity of LLM by replacing one-hot fitting in NTP with flexible target distribution over future tokens.

2. We propose a dynamic weighting scheme based on joint semantic–temporal relevance patterns to determine the importance of future tokens and capture multi-step dependencies.

3. We show that Next-ToBE delivers consistent and tangible gains in terms of language modeling and complex reasoning in both pretraining and fine-tuning over strong MTP baselines.

## 2 RELATED WORK

In recent years, Next-Token Prediction (NTP) has achieved remarkable success in solving difficult problems(Wies et al., 2023), showing powerful learnability (Malach, 2023) and endowing LLMs an

ability to foresee content in the longer future (Wu et al., 2024; Dong et al., 2025; Cencerrado et al., 2025). Meanwhile, critics argue that teacher-forcing strategy overly relies on ground-truth prefixes (Bachmann & Nagarajan, 2024), leading to exposure bias (Schmidt, 2019) and error accumulation (LeCun, 2023) that may hamper long-range reasoning (Bubeck et al., 2023; McCoy et al., 2023). Recently, multiple-token prediction (MTP) architecture (Qi et al., 2020; Gloeckle et al., 2024; Liu et al., 2024a; Chen et al., 2025b; Liu et al., 2025; Samragh et al., 2025) and advanced inference techniques (Cai et al., 2024; Ankner et al., 2024; Cheng et al., 2024) have been proposed to provide richer supervisions during training and mitigate inference error accumulation. Researchers have also proposed transforming training data into a non-causal format to compel the models to capture correlations between non-adjacent target tokens (Bavarian et al., 2022; Thankaraj et al., 2025; Gerontopoulos et al., 2025). Due to space limit, detailed reviews are deferred to the Appendix B.

Different from existing methods, our approach remains within the NTP framework but refines its training objective by replacing the one-hot target vector with a soft target distribution. This can be viewed as a special form of "label smoothing" (Szegedy et al., 2016; Müller et al., 2019), where the one-hot label for the immediate next token is softened by incorporating its temporal and semantic relations with subsequent tokens in the sequence. It requires no modification to LLM architecture and can be easily embedded in to improve LLM anticipatory capacity and reasoning performance.

## 3 METHODOLOGY

We begin by introducing a quantitative measure for token-level anticipatory capacity. A key observation is that LLMs rarely produce strict one-hot vectors. Instead, the model output typically assigns non-zero probabilities to multiple candidate tokens, resembling a form of internal planning. To reflect this, we define the Future-tokens Hit Rate (FtHR), i.e., how well the model prediction at current step $t$, $P_\theta(\mathcal{V}|x_{\leq t}) \in \mathbb{R}^{|\mathcal{V}| \times 1}$, can pre-capture the upcoming $k$ tokens, $\{x_{t+1}, x_{t+2}, ..., x_{t+k}\}$.

$$\text{FtHR}_m^k = \frac{1}{k(N-k-1)} \sum_{t=1}^{N-k-1} \sum_{j=1}^{k} \mathbb{1}\left\{ x_{t+j} \in \text{top}_m\left[ P_\theta(\mathcal{V}|x_{\leq t}) \right] \right\}. \quad (1)$$

Here we consider top-$m$ slots of the LLM output vector, $N$ is the length of the whole sequence, and $\mathcal{V}$ is the vocabulary set. Using this measure, we highlight two key empirical observations:

1. **LLMs possess intrinsic anticipatory behavior by pre-capturing future tokens**. As illustrated in Figure 1a, LLM's prediction at step $t$, $P_\theta(\mathcal{V}|x_{\leq t})$, already contains future tokens, with $\text{FtHR}_{50}^k$-metric ranging from 0.79 to 0.40 when considering windows of length from 2 to 10. These results were obtained using the Qwen-Math-1.5B model on the NuminaMath dataset (Tang et al., 2024). In other words, at each inference step, the LLM does anticipate beyond the immediate next token.

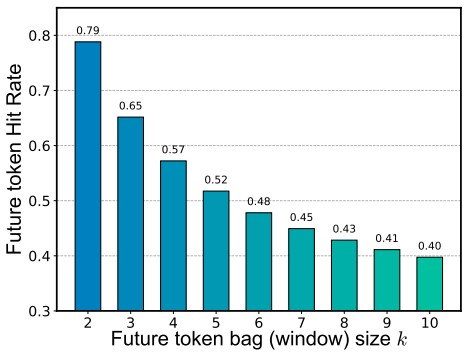
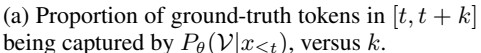
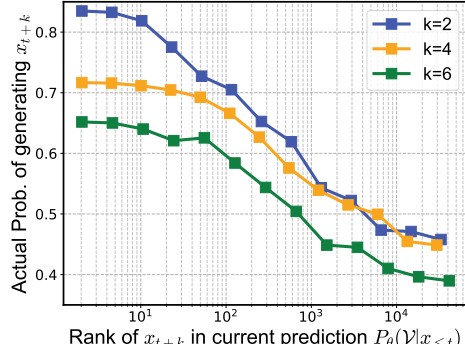

(a) Proportion of ground-truth tokens in $[t, t+k]$ being captured by $P_\theta(\mathcal{V}|x_{\leq t})$, versus $k$.

(b) Prob. of generating $x_{t+k}$ after $k$ steps, vs rank of $x_{t+k}$ within the current prediction $P_\theta(\mathcal{V}|x_{\leq t})$.

Figure 1: The LLM's token-level pre-planning capacity (left), and its correlation with how faithfully an LLM can generate a future token $k$-steps away in an auto-regressive manner (right).

2. **LLM anticipatory capacity is linked to its ability to accurately generate future tokens**. Figure 1b shows that, if a ground-truth token $x_{t+k}$ located $k$ steps ahead receives higher probability

in the current prediction $P_\theta(\mathcal{V}|x_{\leq t})$ (in terms of a higher rank), then $x_{t+k}$ is more likely to be generated (in terms of the probability of auto-regressively generate $x_{t+k}$ at step $t+k$). This suggests that enhancing the anticipatory capacity could practically improve token-level generative accuracies.

Since LLMs possess an intrinsic anticipatory capacity that is positively associated with performance, we ask how this capacity can be more fully activated — or even enhanced — to achieve practical gains. Unfortunately, the one-hot fitting objective forces the model to collapse its predicted probability toward a 'single correct answer', leaving little room for alternative possibilities and forward anticipation. To overcome this, we propose Next-ToBE, which replaces the one-hot objective with a soft-target that effectively integrates broader future information.

## 3.1 PROBLEM FORMULATION

**Next-Token Prediction(NTP):** Given a sequence $\{x_1, x_2, \ldots, x_N\}$ and model $P_\theta$, the teacher-forcing strategy minimizes the following negative log-likelihood at each time step $t$,

$$\mathcal{L}_{\text{NTP}} = - \sum_{t=1}^{N-1} log P_\theta(x_{t+1}|x_{\leq t}), \tag{2}$$

with $P_\theta(\mathcal{V} \mid x_{\leq t})$ the predicted probability distribution over vocabulary $\mathcal{V}$, conditioned on $x_{\leq t}$.

**Multi-Token Prediction (MTP):** Instead of fitting only the next token, MTP make $k$ parallel predictions at time step $t$, each with an auxiliary head $P_{\theta_j}^{(j)}$ for $j = 1, 2, ..., k$ responsible for predicting the ground-truth token $j$ steps away. MTP objective extends the standard NTP loss by minimizing the sum of the negative log-likelihoods of these $k$ future tokens,

$$\mathcal{L}_{\text{MTP}} = - \sum_{t=1}^{N-k-1} \sum_{j=1}^{k} \log P_{\theta_j}^{(j)}(x_{t+j} \mid x_{\leq t}). \tag{3}$$

**Next Token-Bag Exploitation (Next-ToBE):** Next-ToBE replaces the one-hot target distribution in NTP with a carefully designed soft target distribution over the future token bag, thereby effectively activating the anticipatory capacity of LLMs. The loss function is written as

$$\mathcal{L}_{\text{Next-ToBE}} = - \sum_{t=1}^{N-k-1} \left( \underbrace{\log P_\theta(x_{t+1}|x_{\leq t})}_{\textbf{the immediate next token}} + \lambda \underbrace{\sum_{j=2}^{k} w_{x_{t+j}} \log P_\theta(x_{t+j} \mid x_{\leq t})}_{k-1 \textbf{ look-ahead tokens}} \right). \tag{4}$$

The first term is a standard NTP objective, focusing on the **immediate next token** ($x_{t+1}$); while the second term encourages model to fit the subsequent $k-1$ **look-ahead tokens** $\{x_{t+2}, ..., x_{t+k}\}$. The coefficient $\lambda$ balances the contributions of these two terms, and the non-negative weights $w_{x_{t+j}}$ further modulate the relative importance of the $k-1$ look-ahead tokens.

Different from MTP that employs $k$ prediction heads, **Next-ToBE only employs a single head and generates one token at each step in inference**. In this sense, it is in the middle between NTP and MTP: it uses a single head like NTP, but takes into account multiple future tokens at each step like MTP. Interestingly, it is precisely through this one-token-at-a-time generative process that Next-ToBE continuously activates the latent anticipatory capacity of LLMs, thus making more accurate token-level predictions and achieving stronger reasoning performance.

## 3.2 WEIGHTING SCHEME IN TOKEN-BAG EXPLOITATION

We follow the three key principles in Section 1 for designing $\lambda$ and $w_{x_{t+j}}$ in Eq. (4) to effectively capture long-range correlation while maintaining a stable, effective learning process:

1. **Importance of the immediate next token.** The first term in the Next-ToBE loss, which is on fitting the immediate next token, should be assigned dominant importance. This is essential for preserving semantic coherence between the preceding context and the immediate next token. Moreover, the dense supervision of NTP objective is well known to stabilize large-scale optimization

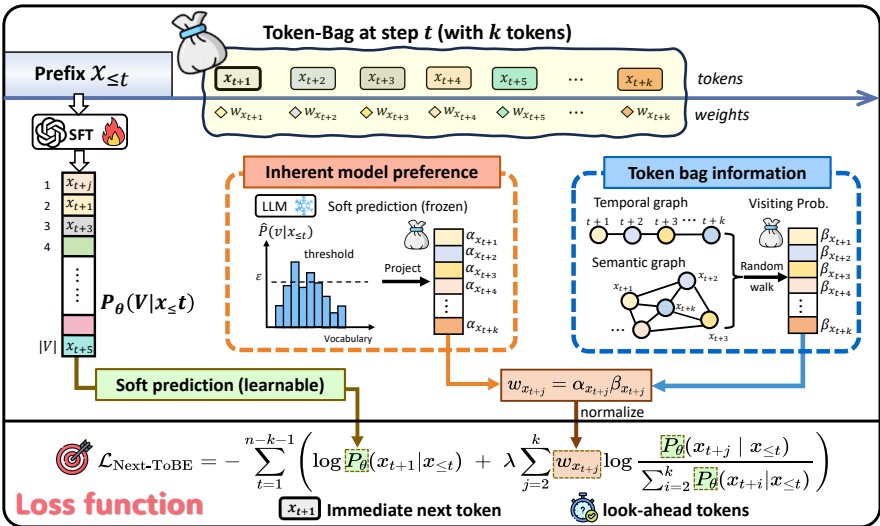

Figure 2: Overview of the Next-ToBE architecture.

and accelerate convergence. To do this, we select a small weighting coefficient $\lambda \in (0,1)$ in Eq. (4), to ensure that the first term dominates. To facilitate choosing $\lambda$, we normalize the second loss term—corresponding to the $k-1$ "look-ahead" tokens—which will be discussed in section 3.3.

The weights $w_{x_{t+j}}$ assigned to future tokens $x_{t+j}$ in Eq. (4) play a crucial role in shaping the learning dynamics. We **decompose the weights into two components**, $w_{x_{t+j}} = \alpha_{x_{t+j}} \beta_{x_{t+j}}$, where $\alpha_{x_{t+j}}$ captures the intrinsic anticipatory preference of the base LLM with respect to the future tokens $w_{x_{t+j}}$, while $\beta_{x_{t+j}}$ encodes their temporal–semantic relevance, as detailed below.

2. **Inherent anticipatory preference of LLM**. The output distribution of an LLM already contains inherent anticipatory preferences that should be fully respected. This enables the model to retain prior knowledge and maintain predictive confidence, especially given the strong capabilities of a base LLM acquired through large-scale pretraining. It also mitigates conflicts between the training signal and model's existing prediction, possibly enhancing training stability and convergence rate.

To achieve this, we introduce a set of weights, $\alpha_{x_{t+j}}$, to track how the LLM's output distribution at the current step $t$ anticipates the $k-1$ look-ahead tokens, $\{x_{t+2}, \ldots, x_{t+k}\}$:

$$\alpha_{x_{t+j}} = \begin{cases} P_\theta(x_{t+j} \mid x_{\leq t})^\gamma, & \text{if } P_\theta(x_{t+j} \mid x_{\leq t}) > \varepsilon \\ 0, & \text{otherwise} \end{cases}, \quad \text{for } j = 2, 3, \ldots, k. \quad (5)$$

Here, $\varepsilon$ is a threshold to filter out low probabilities, and $(\cdot)^\gamma$ ($\gamma = 1/10$) is applied to calibrate remaining probabilities for proper loss re-weighting. Here, $\alpha_{x_{t+j}}$ is frozen in training, i.e., it is obtained from the current "snapshot" of the model at step $t$, and does not contribute to the gradient.

Incorporating $\alpha_{x_{t+j}}$ into future token weights $w_{x_{t+j}}$ forces one to respect LLM's intrinsic anticipatory signal. Specifically, if a future token $x_{t+j}$ is pre-captured by LLM output distribution at the current step, it likely receives higher priority. This can be seen as a form of self-distillation (Hahn & Choi, 2019; Li & Lu, 2021), which helps preserve the knowledge and confidence of the base model.

3. **Semantic/temporal importance of future tokens.** Not all of the $k-1$ look-ahead tokens are equally important. *Temporally*, tokens farther from the current step $t$ may be less relevant and more difficult to predict, so their weights should decay with the temporal gap. *Semantically*, a future token closely related to the immediate next token should have greater importance, as it can enhance local coherence and strengthen the supervision signal by jointly reinforcing the immediate next token. To capture these two preferences, we define the temporal scores and the semantic scores as

$$\tau(t+j) = \exp\left(-j^2/\left(2h^2\right)\right), \quad s(t+j) = \exp\left(\mathbf{x}_{t+1}^\top \mathbf{x}_{t+j}\right), \quad j \in \{2, 3, \ldots, k\}, \quad (6)$$

where $h$ is a bandwidth controlling the temporal decay. These two scores can be combined as

$$\beta_{x_{t+j}} = \tau(t+j)\, s(t+j). \quad (7)$$

However, Eq. (7) ignores the interdependence among the $k-1$ look-ahead tokens. To simultaneously and dynamically account for the relations not only between the immediate next token ($x_{t+1}$) and the look-ahead tokens $\{x_{t+2}, \ldots, x_{t+k}\}$, but also among the look-ahead tokens themselves, we adopt a random-walk–based ranking scheme. Given the token bag $\{x_{t+1}, x_{t+2}, \ldots, x_{t+k}\}$, we construct two $k \times k$ matrices to quantify pairwise temporal and semantic relations:

$$\mathbf{W}_\tau(i,j) = \exp\left(-|i-j|^2/(2h^2)\right), \quad \mathbf{W}_s(i,j) = \exp\left(\mathbf{x}_{t+i}^\top \mathbf{x}_{t+j}\right), \; i,j \in \{1,2,...,k\}, \quad (8)$$

where $\mathbf{x}_i$ is the embedding of token $x_i$ from initial embedding layer of the LLM. Then, both $\mathbf{W}_\tau$ and $\mathbf{W}_s$ are normalized into *transition probability matrices*, such that a random-walk process (Pan et al., 2004; Tong et al., 2008) can be performed on top of them to re-rank the tokens (nodes).

To utilize both relational views, we combine the two matrices by either weighted sum or multiply,

$$\mathbf{W} = \varphi \, \mathbf{W}_\tau + (1-\varphi) \, \mathbf{W}_s, \quad \text{or} \quad \mathbf{W} = \mathbf{W}_\tau \mathbf{W}_s, \quad (9)$$

where $\varphi \in [0,1]$ controls the balance. Both mixing schemes above yield a valid transition matrix - which induces a graph of $k$ nodes/tokens with $\mathbf{W}_{ij}$ encoding their relations (Chen et al., 2025a). We then perform a random walk on $\mathbf{W}$-induced transition graph that starts from the immediate next token $x_{t+1}$, and iteratively transitions to its neighbors, each step with a jump-back probability $\rho$:

$$\boldsymbol{\beta}^{(q+1)} = (1-\rho) \, \mathbf{W} \, \boldsymbol{\beta}^{(q)} + \rho \, \mathbf{e}^{(0)}, \quad (10)$$

where $\boldsymbol{\beta}^{(q)} \in \mathbb{R}^k$ denotes the ranking distribution of the $k$ tokens at iteration $q$, and $\mathbf{e}^{(0)}$ is a one-hot vector indicating the start node $\mathbf{x}_{t+1}$. This iterative process is convergent at

$$\boldsymbol{\beta}^* = \left(\mathbf{I} - (1-\rho)\mathbf{W}\right)^{-1}\mathbf{e}^{(0)}, \quad (11)$$

where $\mathbf{I}$ is identity matrix. The convergent distribution $\boldsymbol{\beta}^*$ jointly accounts for the temporal and semantic relation among the look-ahead tokens. For convenience, we re-write $\beta^*$ in Eq. (11) as

$$\beta_{x_{t+j}} = \boldsymbol{\beta}^*[j]. \quad (12)$$

Empirically, using matrix multiplication in Eq. (9) to construct $\mathbf{W}$ corresponds to the "*Alternate Similarity And Proximity Attention (ASAP-attention)*" (Chen et al., 2025a), which is better than using matrix summation; Besides, weighting schemes based on random-walk scores $\boldsymbol{\beta}^*$ in Eq. (11) is notably better than simple point-wise score fusion in Eq. (7). See Appendix E.3 for details.

We combine (multiply) the LLM-preference based weighting $\alpha_{x_{t+j}}$ in Eq. (5) and temporal-semantic weighting $\beta_{x_{t+j}}$ in Eq. (12), and normalize them to obtain the final weights

$$w_{x_{t+j}} = \frac{\alpha_{x_{t+j}} \beta_{x_{t+j}}}{\sum_{i=2}^{k} \alpha_{x_{t+i}} \beta_{x_{t+i}}}. \quad (13)$$

The normalization ensures that the weights of the look-ahead tokens, $\{w_{x_{t+2}}, \ldots, w_{x_{t+k}}\}$, form a valid probability distribution, which allows for interesting statistical interpretations. Overall, the weights $w_{x_{t+j}}$ integrate both the model's intrinsic capacity and the extrinsic signals present in the data. At step $t$, only those tokens that are simultaneously pre-captured by model output $P_\theta(v|x_{\leq t})$ and actually appearing in the near-future window $[t, t+k]$ will be prioritized as fitting targets.

## 3.3 Normalized Version of Next-ToBE training objective

Although LLMs can pre-capture the look-ahead tokens $\{x_{t+2,...,x_{t+k}}\}$ by giving them non-zero probabilities in the current prediction $P_\theta(\mathcal{V}|x_{\leq t})$, their actual values can be low (i.e., $10^{-7}$), leading to heavily negative log-likelihood. This could dominate the loss and force the learning process to try very hard to raise these small probabilities, which may induce unstable training. To avoid this, we normalize the probabilities $P_\theta(x_{t+j}|x_{\leq t})$ over the look-ahead tokens, as

$$\mathcal{L}_{\text{Next-ToBE}} = -\sum_{t=1}^{n-k-1}\left(\log P_\theta(x_{t+1}|x_{\leq t}) + \lambda \sum_{j=2}^{k} w_{x_{t+j}} \log \frac{P_\theta(x_{t+j} \mid x_{\leq t})}{\sum_{i=2}^{k} P_\theta(x_{t+i}|x_{\leq t})}\right). \quad (14)$$

Empirically, the normalization is important to the performance, as shown in ablation studies in Appendix E.1. Eq. (14) can be seen as the **mix of two cross-entropy losses**: (1) the first is the cross-entropy between the $N$-dimensional LLM output distribution $P_\theta(\mathcal{V}|x_{\leq t})$ and the one-hot distribution encoding the next token $x_{t+1}$; (2) the second is the cross-entropy between the $(k-1)$-dimensional LLM prediction and weight-distribution $\{w_{x_{t+2}}, ..., w_{x_{t+k}}\}$ over look-ahead tokens.

## 4 EXPERIMENTS

Following Mangrulkar et al. (2022), Lu et al. (2023) and Liu et al. (2024b), we adopt the Next-ToBE objective to fine-tune pre-trained LLMs on datasets spanning various reasoning tasks to evaluate the performance gains it can deliver, together with detailed ablation studies.

**Datasets and Baselines.** We used benchmarks from three domains. For mathematical reasoning, following the prior work (Jiao et al., 2024), we randomly sampled 35,000 prompts from the NuminaMath-CoT dataset (Tang et al., 2024) for supervised fine-tuning, without any additional data engineering. For code generation and commonsense reasoning, we used the CodeAlpaca-20k dataset (Wang et al., 2023b) and the Commonsense-15k dataset (Hu et al., 2023) as the training corpus, respectively. We have reported the results of the following competing methods: (1) Next-token prediction (NTP) objective for fine-tuning LLMs; (2) Medusa, the multi-token prediction (MTP) paradigm (Cai et al., 2024), and (3) Mutor (Gerontopoulos et al., 2025), which leverages shared tokens as registers to model future tokens within the NTP framework. Related works that lack code release or cannot be run smoothly were excluded from our side-by-side comparison.

**Training details.** Experiments were run on $4\times$Nvidia RTX A6000. For mathematical reasoning, we used Qwen2.5-Math-1.5B/7B (Yang et al., 2024) and Llama-3.1-8B-Instruct (Dubey et al., 2024) as base models; for code generation and commonsense reasoning, we used Qwen-2.5-1.5/7B and Llama-3.1-8B-instruct. All models were fine-tuned with LoRA adapters (Hu et al., 2022) (rank at 8), and trained for 3 epochs with a batch size of 128, except that for MTP (Cai et al., 2024), an additional one-epoch warm-up stage prior to the full training is used to train the auxiliary prediction heads. The maximum sequence length is 2048 for math and 512 for code generation and commonsense reasoning, for all competing methods. For Next-ToBE, the token bag size is $k = 10$, the threshold $\varepsilon$ in Eq. (5) is $1e - 8$, the bandwidth in Eq. (8) is $h = 2$, the restart probability $\rho$ in Eq. (11) is set to 0.3, and the loss coefficient $\lambda$ in Eq. (14) is 0.05. More detailed settings are in Appendix D.1.

**Evaluation Protocols.** For mathematical reasoning, we have used Qwen-Math (Yang et al., 2024) evaluation codebase, in which models are prompted to reason step by step, and the final answer must be enclosed in ``\boxed{}''. For code generation, we evaluated with official unit-test harnesses (Chen et al., 2021; Austin et al., 2021), reporting pass@1. For commonsense reasoning, we followed official evaluation codebase (Hu et al., 2023). Except for MBPP (Austin et al., 2021) in the code task, which is evaluated under three-shot setting, all other benchmarks are evaluated in zero-shot. Greedy search decoding is used for all methods. Additional details are in Appendix D.2 .

### 4.1 EXPERIMENTAL RESULTS

We empirically investigate the following research questions in this section:

- **Q1**. *Does Next-ToBE improve LLM in anticipating (pre-capturing) future tokens?*
- **Q2**. *Does Next-ToBE improve LLM in accurately generating future tokens?*
- **Q3**. *Does Next-ToBE improve LLM in complex reasoning tasks?*

**Q1** - Figure 3a reports how the LLM pre-captures future tokens by plotting $\text{FtHR}_{50}^{k}$ (the Future-token Hit-Ratio), i.e., how the top-50 slots in the current-step prediction, $P_\theta(\mathcal{V}|x_{\leq t})$, covers future tokens up to $k$ steps, on Qwen-Math-1.5B model. As can be seen, after fine-tuning, the model exhibits a significant increase in the hit ratio across $k \in [2, 10]$, meaning that Next-ToBE effectively enhances the model's anticipatory (pre-planning) capability.

**Q2** - Figure 3b shows the probability that an LLM correctly generates the next $k$ ground-truth tokens auto-regressively from step $t$, conditioned on prefix $x_{\leq t}$. As shown, after fine-tuning, the token-level generative accuracy notably improves across various window sizes. Namely, higher anticipatory capacity indeed turns into higher generative quality, laying the basis for improved reasoning.

**Q3** - Tables 1 and 2 report results on complex reasoning tasks. Across 36 comparisons spanning 3 base model sizes and 12 categories in Mathematics, Code Generation, and Commonsense Reasoning, Next-ToBE attains the highest accuracy in 35 cases, against strong competing methods such as MTP (Gloeckle et al., 2024) and MuToR (Gerontopoulos et al., 2025). These results highlight effectiveness of Next-ToBE in enhancing complex reasoning across diverse tasks and model scales.

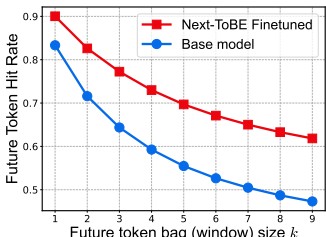
(a) Future-Tokens hit-ratio (antici-pation capacity) vs window size $k$.

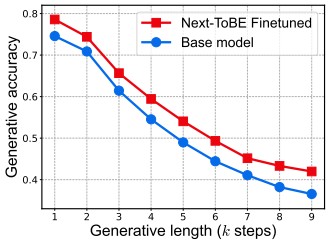
(b) Token-bag generative accuracy versus token-bag size $k$.

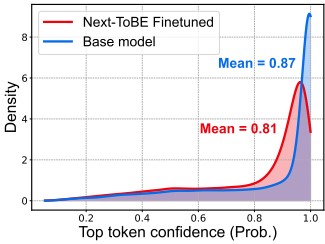
(c) Distribution of top-token prob. after Next-ToBE fine-tuning.

Figure 3: After Next-ToBE fine-tuning, LLM anticipatory capacity (a) and token-level generative accuracy (b) both improve; the next-token confidence (probability) slightly decreases (c).

Table 1: Performance comparisons on five mathematical datasets across three different base LLMs.

| Model | Method | MATH | Minerva Math | Olympiad Bench | AIME24 | AMC23 | Avg. |
|---|---|---|---|---|---|---|---|
| Qwen2.5-Math-1.5B | NTP | 64.8 | 19.1 | 30.7 | 10.0 | 45.0 | 33.9 |
| | MTP | 66.0 | 20.6 | 29.0 | 10.0 | 47.5 | 34.6 |
| | MuToR | 67.6 | 22.1 | 31.4 | 6.7 | 52.5 | 36.1 |
| | Next-ToBE | **68.2** | **23.2** | **31.9** | **13.3** | **55.0** | **38.3** |
| Qwen2.5-Math-7B | NTP | 71.4 | 24.3 | 33.6 | 10.0 | 57.5 | 39.4 |
| | MTP | 72.6 | 23.2 | 32.3 | 13.3 | **60.0** | 40.3 |
| | MuToR | 72.2 | 24.6 | 33.9 | **20.0** | 55.0 | 41.1 |
| | Next-ToBE | **73.6** | **25.4** | **34.8** | **20.0** | **60.0** | **42.7** |
| Llama3.1-8B-instruct | NTP | 48.6 | 21.0 | 14.8 | **10.0** | 25.0 | 23.9 |
| | MTP | 48.2 | 23.5 | 15.6 | 6.7 | 22.5 | 23.3 |
| | MuToR | 49.6 | 23.2 | 15.3 | 6.7 | 27.5 | 24.5 |
| | Next-ToBE | **50.6** | **23.9** | **16.3** | **10.0** | **32.5** | **26.7** |

For Mathematical reasoning (Table 1), Next-ToBE achieves an average accuracy of 38.3%, about 2.2%-4.4% higher than others on Qwen2.5-Math-1.5B, with highest gains on AIME24 (+3.3%) and AMC23 (+2.5%). On Qwen2.5-Math-7B and Llama3.1-8B-instruct, Next-ToBE also attains the highest average scores. For Code generation (Table 2) Next-ToBE has an average accuracy of 55.1% on, being 3.9% and 2.2% higher than MTP and MuToR, respectively, on Qwen2.5-1.5B. When scaling to Qwen2.5-7B, its accuracy rises to 75.2%, outperforming MuToR by 2.7% and NTP by 3.6%, a stable gain as model size grows. Even for Llama3.1-8B-instruct, Next-ToBE delivers an average of 67.3%, exceeding MuToR by 1.9% and NTP by 2.8%. For Commonsense reasoning (Table 2), Next-ToBE achieves an average accuracy of 79.5% on Qwen2.5-1.5B, and rises to 89.4% on Qwen2.5-7B, which is 1.3% - 3.3% higher than competing methods. On Llama3.1-8B-instruct, Next-ToBE has an averaged accuracy of 87.8%, about 1.2%–2.1% higher than other baselines. In addition, we extend our evaluation of Next-ToBE to the larger Qwen3-14B model (Yang et al., 2025). As shown in Appendix E.6, Next-ToBE consistently achieves the highest average accuracy across five mathematical benchmarks. Notably, it delivers absolute gains of 3.4% over MTP and 1.6% over MuToR, exceeding improvements observed on smaller Qwen2.5-Math-7B models. This indicates that Next-ToBE exhibits favorable scalability as model size increases.

**Pretraining setting**. Beyond fine-tuning, we also examined whether Next-ToBE remains effective in pretraining scenario, i.e., without any inherent anticipatory capacity when training from scratch. To this end, we trained a GPT-2 model (124M) (Radford et al., 2019) on WikiText-103 (Merity et al., 2017) using three different objectives: NTP, MTP, and Next-ToBE. Detailed training strategies and evaluation results are provided in Appendix E.7. As shown in Table 9, Next-ToBE improves the Future-token Hit Ratio by 3.29% over NTP and 2.42% over MTP, indicating that anticipatory capacity can indeed be developed from zero. On the downstream HellaSwag QA benchmark, Next-ToBE further achieves absolute gains of +0.86% over NTP and +1.29% over MTP. These results demonstrate that, even in a pure pretraining scenario, Next-ToBE can successfully cultivate anticipatory behaviour from scratch and translate it into improved downstream reasoning performance.

Table 2: Performance comparison on Code Generation and Commonsense Reasoning tasks.

| Model | Method | Code Generation | | | Commonsense Reasoning | | | | | |
|-------|--------|-----------|------|------|------|-----------|-------|-------|------|------|
| | | HumanEval | MBPP | Avg. | PIQA | Hellsawage | ARC-C | ARC-E | SIQA | Avg. |
| Qwen2.5 (1.5B) | NTP | 50.0 | 55.2 | 52.6 | 80.2 | 74.4 | 74.7 | 89.3 | 69.8 | 77.7 |
| | MTP | 48.9 | 53.4 | 51.2 | 78.8 | 74.3 | 71.8 | 87.6 | 68.4 | 76.2 |
| | MuToR | 51.2 | 54.5 | 52.9 | 80.2 | 78.7 | 74.3 | 87.5 | 70.2 | 78.2 |
| | Next-ToBE | **53.7** | **56.4** | **55.1** | **80.7** | **79.3** | **77.1** | **89.9** | **70.4** | **79.5** |
| Qwen2.5 (7B) | NTP | 74.4 | 68.9 | 71.6 | 88.7 | 90.4 | 88.3 | 95.6 | 77.7 | 88.1 |
| | MTP | 76.2 | 69.3 | 72.8 | 88.5 | 90.2 | 87.4 | 95.5 | 77.6 | 87.8 |
| | MuToR | 75.6 | 69.3 | 72.5 | **88.9** | 91.3 | 88.2 | 95.8 | 78 | 88.4 |
| | Next-ToBE | **78.7** | **71.6** | **75.2** | 88.6 | **92.7** | **89.7** | **96.3** | **79.8** | **89.4** |
| Llama3.1 (8B-instruct) | NTP | 65.2 | 63.8 | 64.5 | 86.9 | 87.8 | 83.4 | 93.5 | 76.7 | 85.7 |
| | MTP | 67.1 | 62.3 | 64.7 | 88.2 | 90.1 | 84.8 | 93.4 | 76.7 | 86.6 |
| | MuToR | 67.7 | 63.0 | 65.4 | 87.9 | 90.2 | 84.4 | 93.6 | 74.8 | 86.2 |
| | Next-ToBE | **69.5** | **65.0** | **67.3** | **88.5** | **93.0** | **84.6** | **94.0** | **78.8** | **87.8** |

## 4.2 Hyperparameters Sensitivity

We conduct a series of ablation studies to examine the impact of window size and weighting-related hyperparameters to better understand the robustness and design choices of Next-ToBE.

First, we investigate how the **token-bag size** $k$ affects performance (Appendix E.4). On code generation, Qwen2.5-1.5B and Qwen2.5-7B achieve highest accuracy with $k = 10$, while Llama3.1-8B-instruct peaks at 8. This aligns with expectations: too small windows may fail to provide sufficient future context, whereas overly large windows may introduce noise and confuse the model.

Table 3: Ablation of key hyperparameters on Qwen2.5-Math-1.5B over 5 mathematical benchmarks (averaged).

| $\epsilon(\alpha)$ | Avg. | $h(\beta)$ | Avg. |
|------|------|------|------|
| 1e-4 | 36.6 | 1 | 37.4 |
| 1e-6 | 37.1 | 2 | **38.3** |
| 1e-8 | **38.3** | 3 | 38.0 |
| 1e-10 | 37.6 | 4 | 37.3 |
| 1e-12 | 36.4 | 5 | 37.5 |

We further conduct ablation studies on the hyperparameters related to the weighting scheme, including the threshold $\epsilon$ (for inherent anticipatory preference $\alpha$) and the bandwidth $h$ (for semantic/temporal importance $\beta$), using Qwen2.5-Math-1.5B on five mathematical reasoning benchmarks. The average accuracy results under different settings are summarized in Table 3. Overall, the ablation results demonstrate that the weighting-related hyperparameters in Next-ToBE exhibit stable behavior across a broad range of values. Both the threshold $\epsilon$ and the bandwidth $h$ show smooth performance curves with clear optima, rather than erratic fluctuations. This indicates that Next-ToBE is not overly sensitive to precise hyperparameter settings.

## 4.3 Trade off between Confidence and Reasoning Ability

We examine the impact of **the weight $\lambda$** in Eq. (4), which balances the relative importance of fitting the immediate next token versus the look-ahead tokens. This trade-off plays a critical role in reasoning performance, and also reshapes the confidence dynamics of LLM during generation:

**(1) Confidence dynamics**. In Figure 4a, we evaluate the LLM's confidence by measuring its probability of predicting the next token at each generation step, averaged across the sequence. As shown, increasing $\lambda$ consistently reduces the model's next-token confidence. This happens because the Next-ToBE objective encourages the model to produce smoother predictions at each step by fitting beyond the immediate next token. A similar trend is observed in Figure 3c, where the distribution of the top-token confidence shifts toward a lower mean after Next-ToBE fine-tuning—for example, from 0.87 to 0.81 in mathematical tasks—compared to the base model. We further confirm that Next-ToBE does not introduce distribution drift: KL-divergence analysis over 1,024 step autoregressive generations shows that its token-by-token predictive distribution remains closely aligned with that of a stronger reference model throughout long term generation. (Detailed in Appendix E.8)

**(2) Reasoning performance.** *How does next-token confidence relate to LLM reasoning quality?* As shown in Figure 4b, for code generation and math reasoning tasks with the Qwen1.5B series model, LLM performance improves with increasing $\lambda$ up to a point, after which it declines. This indicates

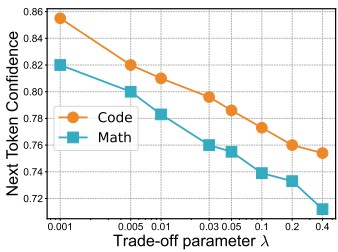 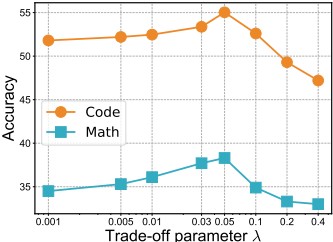 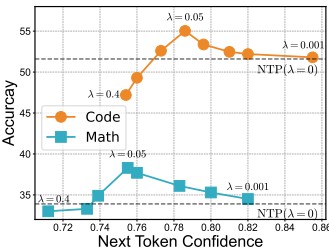

(a) Averaged next-token confidence (probability) vs $\lambda$ in Eq. (4).

(b) Reasoning performance versus $\lambda$ in Eq. (4).

(c) Reasoning performance vs (averaged) next-token confidence.

Figure 4: LLM confidence dynamics and reasoning performance versus trade-off parameter $\lambda$.

that an optimal trade-off exists between fitting the immediate next token and the more distant, look-ahead tokens. Figure 4c plots reasoning accuracy vs next-token confidence: for math, performance peaks around a confidence of 0.755, while in code generation, the optimal confidence is 0.786.

A comprehensive study of LLM confidence dynamics and behavior is a broad research direction, which is related to over-confidence traces regularization (Szegedy et al., 2016), low-confidence traces filtering (Fu et al., 2025), and more others, which we leave for future investigation.

## 4.4 COMPUTATIONAL OVERHEAD

We evaluate runtime and peak memory usage for NTP, MTP, MuToR, and Next-ToBE on the 35K NuminaMath corpus using Qwen2.5-Math-1.5B under identical training settings.

As shown in Table 4, Next-ToBE incurs only a modest overhead relative to standard NTP: +26 minutes of training time ($\sim$20%) and +3.1GB of memory ($\sim$32%). Importantly, it

Table 4: Runtime and memory consumption on mathematical training using Qwen2.5-Math-1.5B.

| Model | Training Time | Peak Memory Consumption |
|---|---|---|
| NTP | 2h 04min | 9,412 MB |
| MTP | 2h 56min | 16,559 MB |
| Mutor | 2h 39min | 39,486 MB |
| Next-ToBE | 2h 30min | 12,552 MB |

remains far more efficient than multi-token prediction baselines, reducing peak memory by up to 68% and training time by up to 15% compared to representative MTP methods. Overall, Next-ToBE delivers substantial performance gains with significantly lower computational cost while retaining the simplicity of a single-head NTP architecture.

Besides, the random-walk weighting module introduces almost no computational burden. Computing its stationary distribution only requires inverting a small $10 \times 10$ matrix per step (from window size $k = 10$), taking roughly $10^{-4}$ seconds in PyTorch. Empirically, this component accounts for approximately about 6.2 minutes of training time (4.1% of a 2.5 hour run on Qwen2.5-Math-1.5B), and its extra memory usage is negligible (34.6 MB out of a 12.6 GB peak).

## 5 CONCLUSION

We presented Next-ToBE, a probabilistic training framework to profile, activate, and exploit the anticipatory capacity of LLMs. Next-ToBE introduces a principled trade-off between fitting the immediate next token and anticipating longer-range future tokens - not via auxiliary prediction heads or architectural changes, but by directly leveraging the inherent probabilistic predictions of LLMs, which we found to be strongly correlated with generative quality. Across a wide range of fine-tuning benchmarks, Next-ToBE consistently outperforms strong multiple-token prediction baselines, with substantial gains in reasoning performance and computational efficiency. Furthermore, anticipatory behavior is also shown to emerge from scratch during Next-ToBE pretraining, highlighting its role as a general and scalable training principle. Finally, we uncover a compelling connection between the probability profiles induced by Next-ToBE and the confidence dynamics of LLMs, offering a new perspective for interpreting and controlling LLM reasoning behavior. We view Next-ToBE as a promising direction for rethinking pretraining across diverse sequential data, including code, time series, and multimodal modeling, where long-horizon anticipation is essential.

ETHICS STATEMENT

This work was conducted in accordance with the ethical standards and code of conduct of ICLR and computer science research community. All experiments were carried out using publicly available datasets, and no personally identifiable or sensitive information was used. We have ensured that our methods, analyses, and results are reported transparently, without fabrication, falsification, or inappropriate manipulation. The authors adhered to principles of academic integrity, fairness, and respect in all aspects of the research and manuscript preparation.

REPRODUCIBILITY

We have made every effort to ensure the reproducibility of our work. All code, scripts, and relevant resources used in this study have been submitted alongside the paper. Detailed instructions regarding model configurations, training procedures, and evaluation protocols are provided in the supplementary materials to facilitate transparent replication of our results.

ACKNOWLEDGEMENTS

We are very grateful to the reviewers for their valuable comments and constructive feedback. This work is supported by the national natural science foundation of China (Grant No. 62276099).

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

# A  USAGE OF LLM

In preparing this manuscript, the authors made only minimal use of a large language model (Chat-GPT). Its role was limited to refining grammar, finding typos and mis-spelled words to improve readability. The study design, experimental procedures, data analysis, interpretation of results, and development of scientific ideas were carried out entirely by the authors without any reliance on the LLM. All substantive intellectual contributions remain solely the work of the authors, who carefully reviewed and verified the final text.

# B  MORE RELATED WORK

We synthesize the disparate arguments related to the next-token prediction paradigm from various research fields to clearly elucidate the nature and contributions of our method, Next-ToBE.

The auto-regressive next-token prediction (NTP) paradigm, which underpins modern language models (Bahdanau et al., 2015; Merrill & Sabharwal, 2024), remains a subject of ongoing debate. Critics argue that teacher-forcing training induces exposure bias—a discrepancy between training and testing conditions wherein the model is exposed to perfect, ground-truth contexts during training but rely on its own potentially flawed generations during inference (Schmidt, 2019; Bachmann & Nagarajan, 2024). This bias may lead to the accumulation and propagation of errors, resulting in substantial deviations from correct solutions (Dziri et al., 2023; LeCun, 2023), thereby compromising the model's capacity for robust, long-range planning (Bubeck et al., 2023; McCoy et al., 2023).

Conversely, proponents highlight NTP's powerful learnability (Malach, 2023), which can transform intractable problems into step-by-step tractable tasks through methods like Chain-of-Thought supervision (Wies et al., 2023). This success in structured reasoning is further supported by a deeper, implicit planning ability (Alabdulmohsin et al., 2024), which has also been demonstrated in recent empirical studies. Probing the internal mechanisms, these studies show that LLMs acquire global information regarding the final response prior to generation (Wu et al., 2024; Dong et al., 2025; Cencerrado et al., 2025). Collectively, these findings indicate that the NTP framework endows LLMs a robust and non-trivial planning capability. This perspective provides the central inspiration for our approach.

To enhance the implicit planning capabilities of NTP models, research has pursued several distinct strategies. A primary approach entails modifying the training data or model architecture to extend the model's predictive horizon. This includes data augmentation techniques that train the model to "fill in the middle" (Bavarian et al., 2022), the incorporation of special tokens to explicitly predict future target tokens (Thankaraj et al., 2025; Gerontopoulos et al., 2025), and architectural modifications that enable bidirectional encoding and prediction (Hu et al., 2025). Although these methods have demonstrated effectiveness, they may also introduce considerable complexity into the data processing and training pipelines.

A second line of research seeks lighter-weight solutions by modifying the training objective itself. For instance, Zuhri et al. (2025) introduced an auxiliary loss to predict the rank of upcoming target tokens without altering the model's architecture. The method we propose advances this lightweight approach. Unlike the aforementioned methods, it requires neither data augmentation nor architectural modifications. Instead, it directly exploits the model's inherent anticipatory signals by manipulating its self-predicted outputs to construct a novel loss target.

Conceptually, our proposed soft "token-bag" target is grounded in a well-established technique: label smoothing (Szegedy et al., 2016). However, instead of smoothing with a uniform distribution over the vocabulary, our method provides a model-informed target that regularizes overconfident predictions—a property shown to improve the generation of diverse and plausible results in early machine translation tasks (Müller et al., 2019; Li & Lu, 2021) and enhance the generalizability and diversity of reasoning results (Li et al., 2025) in the era of LLMs.

Additionally, the "inherent anticipatory signals" manifest as the model's predictive confidences, encoded in the token-level probability distributions that form a reasoning path (Wei et al., 2022; Wang et al., 2023a; Pawitan & Holmes, 2024). The quality of these confidence scores is critical, as they have been shown to strongly correlate with the final answer's correctness (Band et al., 2024; Plaut et al., 2024; Zhao et al., 2025; Jang et al., 2025). The importance of these signals is highlighted by

the effectiveness of various inference-time methods that leverage them, such as modifying prompts to elicit better outputs (Zhao et al., 2024), using decoding strategies like nucleus sampling to filter low-confidence tokens (Holtzman et al., 2020), or aggregating multiple outputs via majority voting (Kang et al., 2025; Fu et al., 2025). Instead of manipulating these signals post-hoc, our training-time method aims to directly improve the quality and calibration of these anticipatory signals at the source to foster more intrinsically robust and reliable reasoning.

Aside from previously introduced advanced NTP learning methods, Multi-Token Prediction (MTP) also enhances the implicit planning capabilities of LLMs (Qi et al., 2020; Gloeckle et al., 2024; Liu et al., 2024a; Chen et al., 2025b; Liu et al., 2025; Samragh et al., 2025). By using auxiliary heads to predict multiple future tokens simultaneously, language models receive richer relationships between tokens during training. Furthermore, advanced inference techniques like speculative decoding reduce generation steps (Cai et al., 2024; Ankner et al., 2024; Cheng et al., 2024), which can mitigate error accumulation and foster creativity (Nagarajan et al., 2025). Although MTP and our method both learn relationship between multiple tokens during training, they ultimately generate single token independently during inference, which limits LLMs' ability to capture the global joint distribution of an entire sequence. This makes exposure bias and the associated problem of cumulative error remain unavoidable.

To overcome the limitations of token-level sequence modeling, researchers have explored paradigms that operate at the sentence level. Non-auto-regressive (NAR) models generate all tokens in parallel, directly capturing the full sentence distribution (Gu et al., 2018; Xiao et al., 2023). Diffusion models refine a complete sequence through a progressive denoising process (Gong et al., 2023; Nie et al., 2025). Both NAR methods ensure consistency between training and testing behaviors, they can generate coherent long texts and even demonstrate potential in reasoning tasks (Gong et al., 2025). Inversely, reliance on a pre-specified output length limits their flexibility that makes auto-regressive NTP models so versatile.

Energy-Based Models (EBMs) provide a unifying framework for text generation by introducing a global energy function that evaluates entire sequences, thereby enforcing global coherence (Dawid & LeCun, 2024). For auto-regressive models, Residual EBMs (Deng et al., 2020) superimpose sentence-level energy scores—either by log-likelihood summation or by training an additional model to compute them—on top of pre-trained models' token-level distributions, mitigating exposure bias and enhancing long-range coherence. Energy-based Diffusion Language Models (Xu et al., 2025) integrate energy functions into each denoising step, often guided by pre-trained auto-regressive models, to correct independence assumptions and improve global consistency. This line of work highlights EBMs as a promising bridge across generative paradigms, effectively combining the local fluency of token-by-token generation with long sentence-level coherence. Consequently, this advancement shows potential in extending the boundaries of reasoning abilities in language models.

## C ANTICIPATORY CAPACITY ANALYSIS

### C.1 FUTURE-TOKEN HIT RATES FOR DIFFERENT TOP-$m$ VALUES AND FUTURE TOKEN-BAG WINDOW SIZES

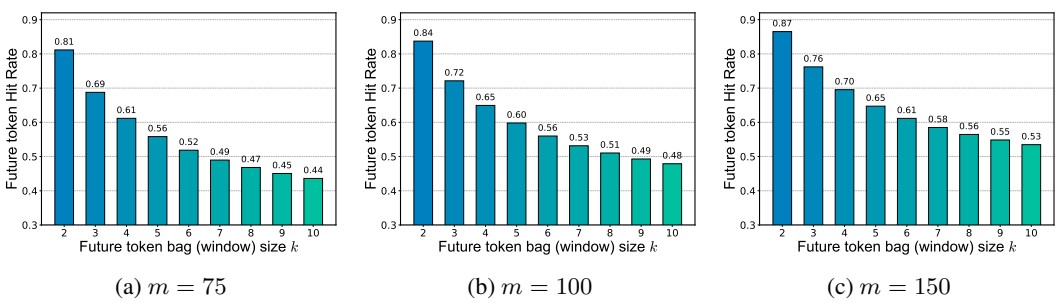

(a) $m = 75$      (b) $m = 100$      (c) $m = 150$

Figure 5: Average Future-tokens Hit Rate (FtHR) under different top-$m$ settings.

To further examine the anticipatory capacity of LLMs, we report the Future-tokens Hit Rate (FtHR) across different values of $m$ (the number of top-ranked tokens considered). Recall that $\text{FtHR}_m^k$ measures how often the ground-truth tokens in the next $k$ steps fall within the top-$m$ predictions of the model at the current step. For this analysis, we randomly sample 100 mathematical reasoning problems from the NuminaMath dataset (Tang et al., 2024).

Figure 5 reports results for three representative values of $m$ (75, 100, and 150). Two consistent trends emerge across all settings: first, the hit rate decreases as the window size $k$ grows, indicating that models are more effective at anticipating near-future tokens; second, larger $m$ values yield higher hit rates, showing that anticipatory signals are spread over a broader set of candidate tokens.

### C.2 DETAILS OF ACTUAL PROBABILITY OF GENERATING WITH RANK IN CURRENT PREDICTION

We further investigate the connection between anticipatory capacity and token-level prediction accuracy. To obtain these results, we consider two complementary evaluation modes:

**Teacher forcing(TF):** At each step $t$, the model is conditioned on the ground truth prefix up to position $t + k - 1$. The probability of target token $x_{t+k}$ is obtained as $P_\theta(x_{t+k}|x_{\leq t+k-1})$. We then pair this ground-truth probability with the rank of $x_{t+k}$ in the prediction distribution at step $t$,forming a sample for analysis. Since all probabilities are computed under the true prefix, this mode represents an idealized setting.

**auto-regressive mode:** The model conditions on the ground-truth prefix up to step $t$, then auto-regressively generates $k - 1$ intermediate tokens before predicting $x_{t+k}$. The probability of the target token is measured as $P_\theta(x_{t+k} \mid x_{\leq t}, \hat{x}_{t+1:t+k-1})$, where $\hat{x}_{t+1:t+k-1}$ are the model's self-generated tokens. This setup reflects a more realistic generation scenario, where prediction accuracy depends not only on the true history but also on the model's own generation quality.

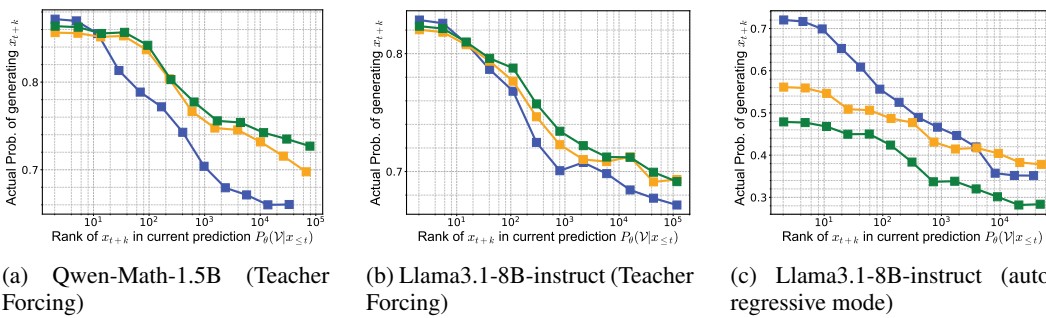

(a)  Qwen-Math-1.5B  (Teacher Forcing)

(b) Llama3.1-8B-instruct (Teacher Forcing)

(c)  Llama3.1-8B-instruct  (auto-regressive mode)

Figure 6: Relations between token level probability and rank across different models and modes

After obtaining the rank of token $x_{t+k}$ in the probability distribution at step $t$ and its actual generating probability $\mathbf{p}_{x_{t+k}}$, we partition the data into numerous logarithmically spaced bins (i.e. 300) based on rank. The mean probability within each bin is then computed to reveal a coarse-grained trend. To further smooth the curve and highlight the overall pattern, we apply Gaussian-weighted averaging to compress these bins into a smaller set of representative points (i.e. 15). Fig. 1b presents the relationship between generating token probability and rank in the auto-regressive setting (Qwen-Math-1.5B). We next present results across different models and modes, as shown in Fig. 6.

## D    TRAINING DETAILS

### D.1   HYPERPARAMETER CONFIGURATIONS

We present the hyperparameter configurations used for Next-ToBE and baseline training methods across all the aforementioned tasks and three pre-trained language models. These detailed settings are provided to facilitate fair comparison and reproducibility. Unless otherwise stated, all experiments are trained for 3 epochs with a global batch size of 128 across all competing methods. The

maximum sequence length is set to 2048 for mathematical reasoning tasks and 512 for code generation and commonsense reasoning tasks.

For Next-ToBE, we adopt the following hyperparameter settings: threshold $\varepsilon = 10^{-8}$ and power scaling factor $\gamma = 0.1$ (Eq. (5)); restart probability $\rho = 0.3$ (Eq. (11)); and a multiplicative fusion of temporal and semantic relations to construct the RWR transition matrix; the bandwidth in Eq. (8) is $h = 2$. In addition, other key parameters such as the future token bag window size $k$ and the loss coefficient $\lambda$ are summarized in Table 5.

Table 5: Best hyperparameter configurations for Next-ToBE on each reasoning task

| Task | Mathematical Reasoning | | | Code Generation | | | Commonsense Reasoning | | |
|---|---|---|---|---|---|---|---|---|---|
| **Model** | Qwen2.5 Math-1.5B | Qwen2.5 Math-7B | Llama3.1 8B-instruct | Qwen2.5 1.5B | Qwen2.5 7B | Llama3.1 8B-instruct | Qwen2.5 1.5B | Qwen2.5 7B | Llama3.1 8B-instruct |
| lr | 3e-5 | 5e-5 | 3e-5 | 5e-5 | 1e-4 | 1e-4 | 1e-4 | 3e-4 | 3e-4 |
| $k$ | 10 | 10 | 9 | 10 | 10 | 8 | 9 | 10 | 10 |
| $\lambda$ | 0.05 | 0.04 | 0.06 | 0.05 | 0.03 | 0.05 | 0.05 | 0.04 | 0.03 |

For NTP, in mathematical reasoning, we set the learning rates to 3e-5, 5e-5 and 3e-5 for Qwen-Math-1.5B, Qwen-Math-7B, and LLaMA-3.1-8B-Instruct, respectively. For code generation, the learning rates for Qwen-1.5B, Qwen2.5-7B, and LLaMA-3.1-8B-Instruct are set to 5e-5, 1e-4, and 3e-4. For commonsense reasoning, the learning rates are set to 1e-4, 3e-4, and 3e-4.

For MTP, we additionally introduce four auxiliary heads, each corresponding to a specific token position, and attach them at the start with random initialization. To alleviate the instability caused by these randomly initialized heads, we freeze all other parameters and pre-train only the auxiliary heads for one epoch before commencing joint training. The learning rate are set to 1e-5, 5e-5 and 5e-5 for mathematical reasoning; 5e-5, 1e-4 and 1e-4 for code generation, and 1e-4, 3e-4 and 3e-4 for commonsense reasoning, corresponding to the Qwen-1.5B series, Qwen-7B series, and LLaMA-3.1-8B-Instruct models, respectively.

For Mutor, the learning rate are set to 3e-5, 5e-5 and 5e-5 for mathematical reasoning; 1e-5, 1e-4 and 1e-4 for code generation, and 1e-4, 3e-4 and 3e-4 for commonsense reasoning, corresponding to the Qwen-1.5B series, Qwen-7B series, and LLaMA-3.1-8B-Instruct models, respectively.

## D.2    PROMPT TEMPLATES

We provide all prompt templates used for training and evaluation.

**Mathematical Reasoning Training and Evaluation Template**

```
<|im_start|>system
Please reason step by step, and put your final answer within \boxed{}.
<|im_end|>
<|im_start|>user
{instruction}<|im_end|>
<|im_start|>assistant
```

> **Zero-shot prompt template for Humaneval and commonsense reasoning evaluation**
>
> ```
> Below is an instruction that describes a task, paired with an input
> that provides further context. Write a response that appropriately
> completes the request.
> ###Instruction:
> {instruction}
> ###Response:
> ```

> **Prompt template for code generation and commonsense reasoning training**
>
> ```
> Below is an instruction that describes a task, paired with an input
> that provides further context. Write a response that appropriately
> completes the request.
> ###Instruction:
> {instruction}
> ###Input:
> {input}
> ###Response:
> ```

> **Three-shot prompt template for MBPP evaluation**
>
> ```
> Below is an instruction that describes a task, paired with an input
> that provides further context. Write a response that appropriately
> completes the request.
> ###Instruction:
> {instruction}
> ###Input:
> {testcase[0]}
> {testcase[1]}
> {testcase[2]}
> ###Response:
> ```

# E MORE EXPERIMENTS

## E.1 IMPACTS OF NORMALIZE TOKEN BAG PROBABILITY

Table 6 summarizes the effect of normalization on token bag distribution in Next-ToBE (see Eq. (14)) during training on Qwen-Math-1.5B for mathematical reasoning. We observe that removing normalization causes a significant drop in overall performance, even below that of the NTP baseline for Olympiad Bench and AIME24 tasks. This degradation arises because, without normalization, small-probability tokens in the token bag can produce disproportionately large losses. As a result, the model overemphasizes distant future tokens while underweighting the immediate next token, ultimately destabilizing training and leading to collapse.

Table 6: Comparison of the Next-ToBE objective with and without normalization across multiple mathematical reasoning benchmarks.

| Method | MATH | MinervaMath | Olympiad Bench | AIME24 | AMC23 | Avg. |
|---|---|---|---|---|---|---|
| NTP | 64.8 | 19.1 | 30.7 | 10.0 | 45.0 | 33.9 |
| Next-ToBE w/o normalize | 63.6 | 20.7 | 25.9 | 6.7 | 45.0 | 32.4 |
| Next-ToBE | **68.2** | **23.1** | **31.9** | **13.3** | **55.0** | **38.3** |

## E.2 DIFFERENT WEIGHTS OF TOKEN BAG INFORMATION

For better analyzing the joint temporal and semantic information, we visualize 30 curves of ranking distributions of with Qwen-Math-1.5B in mathematical tasks, considering multiplication fusion and weighted summation fusion.

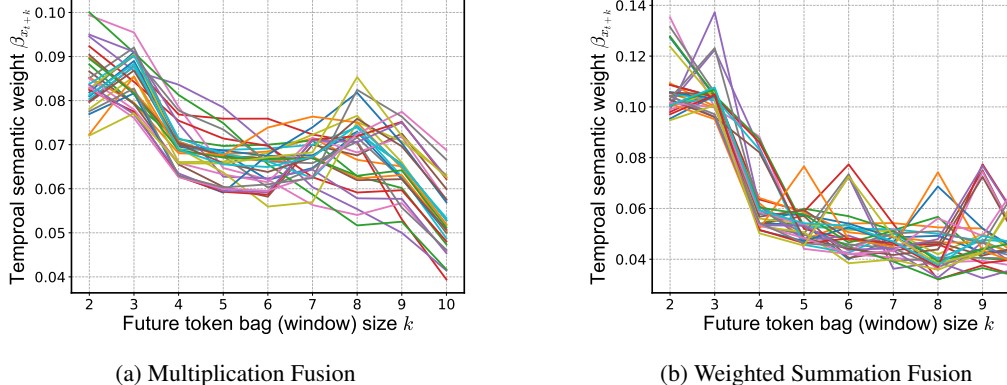

(a) Multiplication Fusion         (b) Weighted Summation Fusion

Figure 7: Examples of the convergent random-walk distribution scores $\beta^*$ plotted against the temporal order of the tokens.

As shown in Fig. 7, both the multiplication and weighted summation fusion methods exhibit an overall decreasing trend as the window size increases. The fluctuations observed, however, indicate that some more distant look-ahead tokens maintain significant semantic correlations with the immediate next token, resulting in higher weights despite their greater temporal distance from the current step. Further, multiplicative fusion (of the temporal proximity matrix $\mathbf{W}_\tau$ and the semantic similarity matrix $\mathbf{W}_s$) leads to generally smoother weights than compared with weighted summation fusion, which we speculate is one of the reasons why the former gives better results.

## E.3 DISCUSSION ABOUT DIFFERENT APPROACHES OF TEMPORAL/SEMANTIC IMPORTANCE OF FUTURE TOKENS

Table 7: Comparison between baseline training methods and Next-ToBE variants with different semantic–temporal joint weighting strategies on five mathematical reasoning benchmarks.

| Method | MATH | Minerva Math | Olympiad Bench | AIME24 | AMC23 | Avg. |
|---|---|---|---|---|---|---|
| **Baselines** | | | | | | |
| NTP | 64.8 | 19.1 | 30.7 | 10.0 | 45.0 | 33.9 |
| MTP | 66.0 | 20.6 | 29.0 | 10.0 | 47.5 | 34.6 |
| MuToR | 67.6 | 22.1 | 31.4 | 6.7 | 52.5 | 36.1 |
| **Next-ToBE Variants** | | | | | | |
| + DM | 68.0 | 22.1 | 31.1 | **13.3** | 52.5 | 37.4 |
| + WSF | **69.2** | 22.8 | 31.7 | 10.0 | **55.0** | 37.8 |
| + MF | 68.2 | **23.1** | **31.9** | 13.3 | **55.0** | **38.3** |

To investigate the impact of different strategies for integrating temporal and semantic information, we design three variants of Next-ToBE and conduct experiments on mathematical reasoning :

(i) Direct Multiplication (DM) combines temporal and semantic weights by element-wise product Eq. (7) without considering interactions between future tokens.

(ii) Weighted Summation Fusion (WSF) combines temporal and semantic matrices with a weighted summation in Eq. (9) (the left equation). In this experiment, we set $\varphi$ to 0.5.

(iii) Multiplication Fusion (MF) combines temporal and semantic information with a multiplication in Eq. (9) (the right equation). We keep this setting for all other experiments.

As shown in Table 7, all Next-ToBE variants outperform the NTP, MTP, and MuToR baselines; even the simplest direct-multiplication scheme (+DM) yields clear average gains (NTP: +3.5%, MTP: +2.8%, MuToR: +1.3%). Among the variants, WSF and MF bring additional improvements, with MF achieving the best overall performance. This suggests that explicitly modeling both semantic similarity and temporal proximity among look-ahead tokens is beneficial. The random-walk formulation was introduced not as a necessity but as a refinement that further enhances performance.

### E.4 IMPACTS OF FUTURE TOKEN BAG WINDOW SIZES

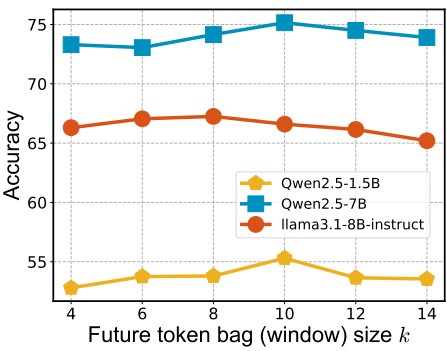

Figure 8: Impacts of future token bag window size

The size of the future window plays a critical role in determining the performance of Next-ToBE. Figure 8 illustrates results from models trained on the CodeAlpaca dataset, reporting average scores on the code generation benchmark with window sizes ranging from 4 to 14 across three models. We observe that Qwen2.5-1.5B and Qwen2.5-7B achieve their highest performance at a window size of 10, while Llama3.1-8B-Instruct reaches its peak at 8. Overall, accuracy tends to improve as the window size increases initially, but declines once the window becomes excessively large.

### E.5 FUTURE-TOKENS HIT RATE DISTRIBUTION

In section 4.1, we have discussed whether Next-ToBE has improved LLM in anticipating future tokens, and found that after fine-tuning with Next-ToBE, the average Future token Hit Ratio on mathematical reasoning across Qwen-Math-1.5B exhibits a significant improvement. For a more intuitive understanding of this improvement, we present the Histogram of Future token Hit Ratio.

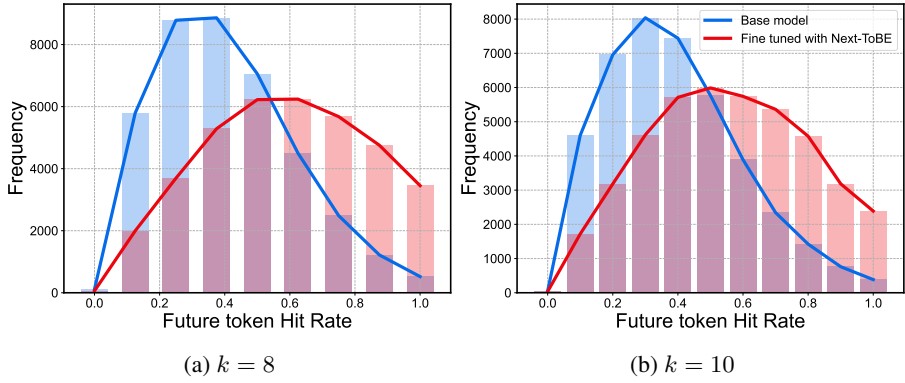

(a) $k = 8$        (b) $k = 10$

Figure 9: Histogram of Future-tokens Hit Rate (FtHR) for different window sizes

As shown in Fig. 9, the distribution of FtHR shifts rightward after Next-ToBE fine-tuning, compared to the base model. This indicates that fine-tuning lea relative high range, thereby demonstrating a clear improvement in anticipatory capability.

## E.6 BEHAVIOR ON LARGER MODEL

To assess whether the benefits of Next-ToBE persist as the model scale increases, we extend our study from 8B models to a larger 14B-parameter foundation model (Qwen3-14B). We fine-tuned the 14B model using Next-ToBE and compared it against three baselines (NTP, MTP, and MuToR). The average accuracies on five mathematical reasoning benchmarks are reported in Table 8.

Table 8: Performance comparisons of different competing methods on five mathematical datasets using Qwen3-14B.

| Method | Math | Minerva Math | Olympiad Bench | AIME24 | AMC23 | Avg. |
|---|---|---|---|---|---|---|
| NTP | 64.2 | 29.4 | 24.0 | 3.3 | 40.0 | 32.2 |
| MTP | 64.6 | 30.1 | 24.3 | 3.3 | 35.0 | 31.5 |
| MuToR | 64.0 | 30.1 | 23.4 | **6.7** | 42.5 | 33.3 |
| Next-ToBE | **65.4** | **30.5** | **24.6** | **6.7** | **47.5** | **34.9** |

As shown in Table 8, Next-ToBE continues to achieve the highest performance across all benchmarks on the 14B model, with absolute gains of 3.4% over MTP, 2.7% over NTP, and 1.6% over MuToR. These improvements are comparable or even larger than those observed on the smaller 8B model, demonstrating Next-ToBE exhibits strong scalability and remains effective on larger models.

## E.7 PRE-TRAINING SETTING

Rather than restricting our study to the post-training scenario, we further explore the applicability of Next-ToBE in the pre-training stage. Below, we describe a detailed setup (with necessary adjustments for the pretraining scenario) and report results.

**Model and Data.** We pretrained a GPT-2–style model (124M) from scratch on 100k samples randomly drawn from the training split of the WikiText-103 dataset, for 10 epochs.

**Training Strategy.** The trade-off parameter $\lambda$ in Eq.14 was increased from 0 to 0.1 over 10 epochs following a quadratic schedule; look-ahead token weights followed the simple scheme in Eq.7, and the weight scheme over intrinsic anticipatory signals in Eq.5 was removed (as the model trained from scratch has no such capacity).

**Evaluation metrics.** (1) perplexity (PPL) of the generated text; (2) Future-token Hit Ratio (FtHR), which reflects anticipatory capacity; (3) Accuracy on the QA task. Metrics (1) and (2) are based on the Wikitext-103 test-split, metric (3) is computed on the Hellaswag test-set.

Table 9: Comparison of NTP, MTP, and Next-ToBE under pre-training settings for a GPT-2 model.

| Method | PPL $\downarrow$ | FtHR$_{50}^{10}$ $\uparrow$ | HellaSwag Acc $\uparrow$ |
|---|---|---|---|
| NTP | **28.67** | 22.51 | 25.27 |
| MTP | 73.60 | 23.38 | 24.84 |
| Next-ToBE | 34.32 | **25.80** | **26.13** |

As shown in Table 9 ,the following observations can be made:

(1) **Anticipatory capacity.** Next-ToBE shows a notably higher anticipatory capability against NTP and MTP (+2.4-3.3 FtHR, 10-15% higher). Given that the model is trained from scratch with no anticipatory capacity at all, these results provide compelling evidence that Next-ToBE not only enhances the anticipatory capacity by fine-tuning a pretrained LLM but effectively develop such a capability from zero through pretraining, surpassing both NTP and MTP with the same training condition.

(2) **Perplexity.** Next-ToBE exhibits a higher perplexity than NTP (34.3 vs. 28.7). This is entirely expected because Next-ToBE was designed exactly to trade the short-term, next-token-prediction certainty (as measured by PPL) for improved longer-range anticipation/reasoning. This has been supported by gains in sequence-level generation accuracy (Fig. 3b) and performance on complex reasoning tasks (Table 1 & 2). In comparison, MTP improves FtHR by only 3% over NTP - far

below our 15% gain - while incurring a much larger PPL penalty (73.6, 2.15× ours). These results indicate that Next-ToBE achieves a more efficient balance between local token prediction and global sequence modeling.

(3) **QA-Tasks.** On the HellaSwag commonsense QA benchmark, Next-ToBE improves accuracy by +0.86% over NTP and +1.29% over MTP, indicating that its enhanced anticipatory behavior indeed translates into better reasoning performance. Though our experiments were small in scale, the comparisons are strictly controlled, with all methods trained on exactly the same data for the same number of epochs. So the higher QA-task accuracy of Next-ToBE provides clear evidence of its advantage.

### E.8 DISTRIBUTIONAL DRIFT ANALYSIS

We investigate whether fine-tuning with Next-ToBE leads to distributional drift during long-horizon autoregressive generation. To assess this, we generate 1,024-token sequences using diverse Math-benchmark prompts (test set) and compare, at each decoding step $t$, the next-token distribution of our model (Qwen2.5-Math-7B fine-tuned with Next-ToBE) against a stronger reference model (Qwen2.5-Math-PRM-7B) using KL divergence. Averaged over 100 sequences, the KL divergence at final step ($t = 1024$) is 0.15 for KL(ours|ref) and 0.17 for KL(ref|ours), and the mean KL across all 1024 steps is 0.19 and 0.26, respectively. These small and stable divergences indicate that Next-ToBE preserves the training-time token distribution throughout free-running generation and does not exhibit long-term drift.

