# OpenReview forum: "Next-ToBE: Probabilistic Next Token-Bag Exploitation for Activating Anticipatory Capacity in LLMs"
_ICLR.cc/2026/Conference — ICLR 2026 Poster_

### Official Review · Reviewer_f6SD · 2025-10-31

**Soundness:** 3
**Presentation:** 4
**Contribution:** 3
**Rating:** 4
**Confidence:** 3

**Summary:**

This paper proposes Next-ToBE, a new fine-tuning objective as an alternative to vanilla NTP.

The paper points out that regular LLMs only focus on predicting the very next word/token, which stops them from actually thinking ahead (but it does already exist to a certain extent).

The authors created their new training method, Next-ToBE that teaches the model to predict not just the next token, but also a few tokens down the line at the same time. Instead of simply predicting the next work, it predicts a probability of the next few words.

The key idea is that this helps the model develop better planning skills by making it look further ahead during training.

The authors demonstrate Next-ToBE's effectiveness by fine-tuning Qwen and Llama models on tasks in mathematical, code, and commonsense reasoning.

**Strengths:**

- The paper is well written, motivated, an executed
- The empirical results (Table 1) look solid. In the ran experiments, Next-ToBE is clearly the best method.
- The proposed Future-tokens Hit Rate (FtHR) is a nice metric
- I appreciate the detail in training method, data, eval pipeline

**Weaknesses:**

- My biggest concern is that it's only done on models that are already post-trained. Pure NTP/MTP is typically done for pretraining, so I'm not sure why this is done on post-trained models. I think the results would be a lot more convincing if it was done on base models, on pretraining data, with full model tuning (not just lora).
- The conclusion claims that it is "simple to implement", the method is quite complicated, especially compared to MTP. Particularly the part about the random walk on the graph. In its current form, it will likely not be adopted by many.
- A lot of the method is heuristically driven. E.g. chosen values of lambda. How can others use this method? Will it always work out of the box on any model with the fixed hyperparams? I think more ablations of the hyperparams (alpha, beta, lambda) are necessary
- It's unclear how much more efficient Next-ToBE is than other methods. There is overhead from computing the values for the lookahead tokens, but not sure how much.

**Questions:**

- Why are all experiments/results shown on post-trained models? Why not evaluate this on a base model? Do the same properties hold there? I would consider raising my score if this can be answered.
- How expensive is the random-walk weighting calculation?

---

> ### Author Response · Authors · 2025-11-21
> **Response to Reviewer f6SD (Part 1)**
>
> We sincerely appreciate the reviewer's valuable comments, which are extremely helpful and constructive in deepening our understanding and enhancing the completeness of our work. Below, we carefully addressed each point and hope our responses meet your expectations.
>
> **Response to W1&Q1 - "Only worked with post-trained models, should also work on full model tuning/pretraining"**
>
> We highly appreciate the reviewer’s insightful comment. In the previous version, we focused on post-training because:
> 1. Recent baselines like MuToR and Medusa were evaluated in fine-tuning scenarios;
> 2. Our method was originally motivated by activating/enhancing anticipatory capacity inherent **in pretrained LLMs** (Fig. 1, Fig. 3), making post-training a natural initial testbed;
> 3. Full-scale pretraining can be prohibitively expensive for us.
>
> We fully agree with the reviewer that full-model pretraining is an important scenario, so we conducted new experiments in this setting. Below we describe detailed setup (with necessary adjustments for the pretraining scenario), and report results.
>
> **Model and Data.** We pretrained a GPT-2–style model (124M) from scratch on 100k samples randomly drawn from the WikiText-103 training split, for 10 epochs.
>
> **Training Strategy.** The trade-off parameter λ in Eq. (14) was increased from 0 to 0.1 over 10 epochs following a quadratic schedule; look-ahead token weights followed the simple scheme in Eq. (7), and the intrinsic anticipatory signal in Eq. (5) was removed (model trained from scratch has no such capacity).
>
> **Evaluation metrics:** (1) perplexity (PPL); (2) Future-token Hit Ratio (FtHR), reflective of anticipatory capacity; (3) Accuracy in QA. (1) and (2) were based on Wikitext-103 test-split, and (3) on Hellaswag test-set.
>
> | Method   | PPL ↓    | $\mathrm{FtHR}_{50}^{10}$ ↑ |Hellaswag Acc↑ |
> | ------------- | :---------: | :---------------------:|:-------------: |
> | NTP     | **28.67**     | 22.51    | 25.27 |
> | MTP     | 73.60     | 23.38    | 24.84 |
> | Next-ToBE | 34.32    | **25.80**    | **26.13** |
>
> The following observations can be made:
> 1. **Anticipatory capacity.** Next-ToBE shows notably higher anticipatory capability against NTP and MTP (+2.4-3.3 FtHR, \~10-15% higher). Given that the model is trained from scratch with no anticipatory capacity at all, these results provide compelling evidence that Next-ToBE not only enhances the anticipatory capacity by fine-tuning a pretrained LLM,**but can also effectively develop such capability from zero through pretraining,** surpassing both NTP and MTP with same training condition.
> 2. **Perplexity.** Next-ToBE exhibits a higher perplexity than NTP (34.3 vs. 28.7). This is entirely expected because Next-ToBE was designed exactly to trade the short-term, next-token-prediction certainty (as measured by PPL) for improved longer-range anticipation/reasoning. This has been supported by gains in sequence-level generation accuracy (Fig. 3b) and performance on complex reasoning tasks (Table 1 & 2). In comparison, **MTP improves FtHR by only ~3% over NTP - far below our 15% gain - while incurring a much larger PPL penalty (73.6, ~2.15× ours)**. Namely, Next-ToBE had a more efficient and balanced trade-off between local token prediction and global sequence modeling.
> 3. **QA-Task.** On the HellaSwag commonsense QA benchmark, Next-ToBE improves accuracy by +0.86% over NTP and +1.29% over MTP, indicating that its enhanced anticipatory behavior indeed translates into better reasoning performance. Though our experiments were small in scale, the comparisons are strictly controlled, with all methods trained on exactly the same data for the same number of epochs. So the higher QA-task accuracy of Next-ToBE provides clear evidence of its advantage.
>
> Due to limited GPUs and needs to obtain results quickly for many parallel runs (different methods and tasks), we were only able to work with small models and limited corpora. Nevertheless, they provide clear and consistent evidence of Next-ToBE’s advantage in the pretraining scenario, in line with our expectations. We will conduct more thorough evaluations going forward, and we are happy to run experiments for any specific settings upon your request. Again, we thank the reviewer for your constructive suggestions.

---

> > ### Comment · Reviewer_f6SD · 2025-11-23
> >
> > Thanks for the clarifications. I'm raising my score from a 4 to a 6.

---

> ### Author Response · Authors · 2025-11-21
> **Response to Reviewer f6SD (Part 2)**
>
> **Response to W2 - "Why the method is simple to implement, especially compared to MTP"**
>
> We thank the reviewer for this important comment and we apologize for the lack of clarity in our presentations. Our clarifications are provided below:
> 1. **No architectural modifications required.** Next-ToBE uses the same training and inference architecture as standard LLMs with a single output head. In contrast, MTP relies on four auxiliary heads and an additional warm-up stage to stabilize training, introducing nontrivial code changes and higher memory and training costs. From this angle, Next-ToBE is simpler and more straightforward.
> 2. **Compact, low-overhead random-walk module.** The random-walk module, based primarily on small matrix inversions, is a compact plug-in (\~60 lines) that reweights look-ahead tokens for the auxiliary loss. It is cleanly decoupled from the rest of the model and adds only \~4% runtime overhead. In fact, even a simpler weighting scheme (\~15 lines) still provides clear gains over the baselines, as we reported in Appendix E.3. We will provide clean, ready-to-use implementation options to enhance usability.
>
> **Response to W3 - "More ablations of the hyperparameters"**
>
> We thank the reviewer for this important comment, and our responses are listed below.
>
> 1. We highly appreciate the reviewer’s broader point regarding engineering complexity in LLM systems. To address this, we will provide clear user guidelines in our code release, including default hyperparameter configurations and recommended grid points for validation-based tuning, to ensure our code is both easy to use and robust.
>
> 2. In practice, the hyperparameters of Next-ToBE were easy to set. Across all tasks (code, math, and common-sense QA) and models (Qwen, Llama), we fixed ε = 1e-8 (threshold), and h = 2 (Gaussian width), and only selected λ (trade-off parameter) and k (window size) from a few candidates (grid points) based on small validation-set.
>
> 3. Based on your suggestion, we conducted ablations on Qwen2.5-Math-1.5B and report avg. accuracy over five mathematical reasoning benchmarks. As shown, the Next-ToBE performance curve w.r.t. hyperparameters exhibits a simple, well-defined peak, indicating relatively low sensitivity and ease of parameter selection in practical application scenarios.
>
> | $\epsilon$ ($\alpha$)  |  ACC.  |	 $\lambda$  |  ACC.  |  h ($\beta$)  |  ACC.  |
> |:----------------:|:--------------:|:--------------:|:-------------:|:--------:|:-----------:|
> |  1e-4		 |  36.6	   |	 	0.01  | 	  37.1	 |  1  |  37.4  |
> |  1e-6		 |  37.1	   |	 	0.03  | 	  37.7	 |  2  |  **38.3**  |
> |  1e-8		 |  **38.3**	   |	 	0.05  | 	  **38.3**	 |  3  |  38.0  |
> |  1e-10		 |  37.6	   |	 	0.07  | 	  36.9	 |  4  |  37.3  |
> |  1e-12		 |  36.4	   |	 	0.09  | 	  35.5	 |  5  |  37.5  |
>
>
> **Response to W4 - "Computational efficiency/overhead of the method"**
>
> We thank the reviewer for this detailed comment. To quantify the computational overhead of Next-ToBE, we measured training time and peak memory usage on Qwen2.5-Math-1.5B for the math-reasoning:
>
> | Method   | 	Training Time   |  Peak Memory usage  |
> | ------------- | :-------------------------: |:-----------------------------: |
> | NTP      |    **2 h 4 min**		|    **9412.19MB** 		|
> | MTP      |    2 h 56 min		|    16559.13MB 		|
> | Mutor      |    2 h 39 min	|    39486.42MB 		|
> | Next-ToBE   |    2 h 30 min	|    12552.55MB 		|
>
> As observed, Next-ToBE incurs a modest overhead relative to standard NTP architecture, i.e., **26 more minutes (+20%)  and 3GB more memory (+32%)**. However, it remains substantially more efficient than other baselines: MTP requires **\~41% more training time and \~75% more memory** against NTP. while MuToR requires **28% more time and 320% more memory (\~39 GB vs. \~9 GB)** against NTP. These results show that Next-ToBE achieves its improvements with lower cost.
>
> **Response to Q2 - "Cost of the random-walk component"**
>
> The random-walk weighting is cheap. Its stationary distribution requires inverting a 10×10 matrix (from window size k = 10) for each step, taking \~10⁻⁴ sec. in PyTorch. Empirically, random-walk module took 6.2 minutes - about **4%** of the total 2.5-hour training time on Qwen2.5-Math-1.5B. Its memory overhead is negligible (**30 MB** out of a **12.6 GB** peak).

---

> ### Author Response · Authors · 2025-11-24
> **Sincere Appreciation for Your Review and Guidance**
>
> Dear Reviewer,
>
> We would like to express our sincere gratitude once again for your valuable time and constructive comments, which enabled us to significantly improve the quality of our work. Thank you for the support, insight, and professionalism you have brought to the review process and to the community.
>
> All Authors

---

### Official Review · Reviewer_ianY · 2025-11-01

**Soundness:** 4
**Presentation:** 3
**Contribution:** 4
**Rating:** 8
**Confidence:** 3

**Summary:**

This paper provides a loss function that encourages an autoregressive LM’s prediction to include more information about future tokens, without changing the underlying Transformer architecture or compromising inference accuracy or speed. The idea is to use a **mixture of two cross-entropy losses**: (i) between the LM head and the correct **next** token (one-hot), and (ii) between the model’s **bag-renormalized** prediction over the next (k) tokens and a set of importance weights over those tokens. Next-ToBE **nearly unanimously** outperforms NTP, MTP, and MuToR across numerous math, code, and commonsense reasoning benchmarks. Since there is only one LM output distribution per position, inference speed stays comparable to NTP (unlike MTP). The appendix contains many additional informative experiments.

**Strengths:**

Clearly written, strong and extensively evaluated results, and an improvement on a central topic (raw language modeling itself).

**Weaknesses:**

The weighting scheme seems ad hoc and complicated, containing several hyperparameters (lambda, k, epsilon, gamma, h, rho) and the fusion of W_tau and W_s). However, the random-walk matrix is backed up by ablations, and upon reflection, the random walk here may not be so ad hoc.
Missing some minor distributional diagnostics (see questions).

**Questions:**

1. How many tokens of CoT were used in the math evaluation protocols (a maximum is provided, but not mean/SD)? I would like to understand why autoregressive sampling does not appear to pull the Next-ToBE model off the token-by-token distribution. Conceptually, since the loss divides by the probability mass assigned to the future (k)-bag, the auxiliary term should mainly reshuffle probabilities **within** that set, but I would still expect the opposite effect from what Figure 3b shows.
2. Does long-term Next-ToBE **autoregressive** sampling diverge from the training distribution—e.g., as measured by KL **to/from** a stronger language model?
3. Does this technique still work if you train from scratch rather than fine-tuning a pretrained baseline? I ask in part because the alpha weights focus on tokens that already receive probability mass, which will start **at random** when training from scratch.

---

> ### Author Response · Authors · 2025-11-21
> **Response to Reviewer ianY (Part 1)**
>
> We are deeply grateful for the reviewer's insightful and encouraging comments. We have learned a great deal, particularly from your constructive feedback and your inspiring, broad perspectives. We address each point in detail below.
>
> **Response to Weakness:** We thank the reviewer for this thoughtful comment. We fully agree that our presentations can be further simplified. We highly appreciate your comment that “the random walk may not be so ad hoc” in light of the ablation results. In fact, our initial simple weighting scheme in Eq. (7) already yielded clear gains over baselines (NTP: +3.5%, MTP: +2.8%, MuToR: +1.3%) in Appendix E.3. The random-walk module was added to further improve the result, and we will clarify its motivation and design in the revision.
>
> **Response to Q1 - "CoT length, why Next-ToBE can preserve token-by-token distributions"**
>
> We thank the reviewer for this insightful comment. In our evaluations, the CoT lengths for math reasoning tasks have a mean of 867 tokens with a standard deviation of 275, for code generation tasks the mean length is 180 tokens with a standard deviation of 71. We will include these statistics in the revision.
>
> Regarding whether Next-ToBE preserves token-by-token distribution during autoregressive generation, we believe this largely holds for three reasons:
> 1. **Dominance of next-token supervision.** The auxiliary loss redistributes only a small portion of the target probability mass (weight\~0.05) to the k look-ahead tokens, while the immediate next token still dominates (weight\~1). Thus, autoregressive sampling largely preserves local next-token predictions.
> 2. **Propagation of lookahead consistency.** As generation unfolds, lookahead tokens gradually become next tokens, allowing their influence to propagate smoothly along the sequence, thus improving both sequence-level generative accuracy (Fig. 3b) and reasoning performance (Table 1 & 2).
> 3. **Argmax decoding.** Argmax sampling might further stabilize the generated distribution by suppressing small fluctuations in predicted probabilities.
>
> Finally, as discussed in our response to Q2, we provide additional empirical evidence -based on your suggestion - that free-running Next-ToBE generations (1,024 token length) remain stable and show no noticeable drift from the training-time distribution. We sincerely appreciate your insightful and inspiring comment.
>
> **Response to Q2 - "Long-term Next-ToBE autoregressive sampling vs the training distribution"**
>
> We truly appreciate this insightful comment regarding whether free-running Next-ToBE may drift from the training distribution. Based on your suggestion, we autoregressively generated 1,024-token sequences with Next-ToBE using diverse prompts from the Math benchmark (test-set). At each decoding step t, we queried a stronger reference LM to obtain its next-token distribution and computed the KL divergence across the trajectory - from step 1 to t - between the two models. To ensure vocabulary consistency, we used **Qwen2.5-Math-7B (fine-tuned with Next-ToBE)** as our model and **Qwen2.5-Math-PRM-7B (official RL-fine-tuned model)** as the stronger reference. Results were averaged over 100 sequences. Our observations are:
> 1. **Low KL divergence across long sequences.** The average KL at the final position (t=1024) is 0.15 for $\text{KL}(\text{ours} || \text{ref})$ and 0.17 for $\text{KL}(\text{ref} || \text{ours})$. When averaging across all 1,024 steps, the values are 0.19 and 0.26, respectively. These low magnitudes indicate that Next-ToBE preserves the token-by-token training-time distribution even under long free-running generation.
> 2. **Stable KL curves over time.** The KL-divergence curve remains flat across decoding steps, with only minor fluctuations uniformly distributed across the sequences, suggesting a stable long-horizon autoregressive behavior without noticeable drift.

---

> ### Author Response · Authors · 2025-11-21
> **Response to Reviewer ianY (Part 2)**
>
> **Response to Q3 - "Whether the method still works when training a model from scratch?"**
>
> We thank the reviewer for this insightful angle. Indeed, our technique remains effective even when pre-training a model from scratch. To show this, we pre-trained a **GPT-2–style model** (124M) from scratch on **WikiText-103** using three different objectives (NTP, MTP, and Next-ToBE) with the same number of epochs and compared their performance. Because such a model has no built-in anticipatory capability at initialization, we removed the inner anticipation signal (α-weights in Eq. (5)) and used only the simple weighting scheme (β-weights in Eq. (7)). The balancing parameter λ increased from 0 to 0.1 over the epochs following a quadratic schedule.
>
> We evaluated the model using three metrics: perplexity (PPL), Future-token Hit Ratio (FtHR), and HellaSwag accuracy for QA tasks. PPL and FtHR were computed on the Wikitext-103 test split, while HellaSwag accuracy was measured on the test split of the HellaSwag CommonsenseQA dataset.
>
> | Method   |  $\mathrm{FtHR}_{50}^{10}$ ↑ | Hellaswag Acc↑ |  PPL ↓  |
> | ------------- | :---------: | :---------------------: |:-------------: |
> | NTP     |  22.51    | 25.27    | **28.67** |
> | MTP     |  23.38    |  24.84   |  73.60 |
> | Next-ToBE | **25.80** |  **26.13**  |  34.32  |
>
> The following observations can be made:
> 1. **Anticipatory capacity.** Next-ToBE shows notably higher anticipatory capability against NTP and MTP (+2.4-3.3 FtHR, \~10-15% higher). Given that the model is trained from scratch with no anticipatory capacity at all, these results provide compelling evidence that Next-ToBE not only enhances the anticipatory capacity by fine-tuning a pretrained LLM, **but can also effectively develop such capability from zero through pretraining**, surpassing both NTP and MTP with the same training condition.
> 2. **QA-Accuracy.** On HellaSwag commonsense QA, Next-ToBE improves accuracy by +0.86% over NTP and +1.29% over MTP, indicating that its enhanced anticipatory behavior indeed translates into better reasoning performance. Though our experiments were small in scale, the comparisons are strictly controlled, with all methods trained on exactly the same data for the same number of epochs. So the higher QA-task accuracy of Next-ToBE provides clear evidence of its advantage.
> 3. **Perplexity.** Next-ToBE exhibits a higher perplexity than NTP (34.3 vs. 28.7). This is entirely expected because Next-ToBE was designed exactly to trade the short-term, next-token-prediction certainty (as measured by PPL) for improved longer-range anticipation and reasoning. This has been supported by gains in sequence-level generation accuracy (Fig. 3b) and performance on complex reasoning tasks (Table 1 & 2). In comparison, **MTP improves FtHR by only \~3% over NTP - far below our 15% gain - while incurring a much larger PPL cost (73.6, \~2.15× ours)**. In other words, Next-ToBE achieved a more efficient and balanced trade-off between local token prediction and global sequence modeling.

---

### Official Review · Reviewer_9ScA · 2025-11-02

**Soundness:** 3
**Presentation:** 3
**Contribution:** 3
**Rating:** 6
**Confidence:** 3

**Summary:**

The paper proposes Next-ToBE, a training objective to improve the anticipatory capability in LLMs. Next-ToBE modifies the standard next-token prediction objective in LLM training by replacing the one-hot target vector with a soft distribution over additional future tokens. The model incorporates a weighting scheme that assigns a dominant weight for the immediate next token, preserves the model's own anticipatory preferences, and dynamically adjusts future token weights based on temporal and semantic importance. Experiments on diverse reasoning settings (math reasoning, code generation, and common-sense reasoning) using Qwen and Llama models show consistent accuracy improvement over NTP and MTP baselines. The approach requires no architectural change and reduces peak memory usage compared to MTP methods.

**Strengths:**

- The method is well-motivated by the two key empirical observations. The defined future token hit rate metric and the corresponding analysis provide valuable insights into LLMs' intrinsic anticipatory behavior and how it's linked to generative accuracies.
- The objective is straightforward, and the design choices are well-motivated and supported by theoretical justification and ablation studies.
- The method is simple and requires no architectural or inference changes. It reduces peak memory usage compared to MTP methods.
- The paper provides extensive experiments across diverse reasoning tasks and model families and demonstrates consistent improvements.

**Weaknesses:**

- The scope of baselines could be limited. The related work section acknowledges connections to label smoothing and other MTP-related baselines, but they are not evaluated as baselines, including ProphetNet, token order prediction, and label smoothing.
- Direct quantitative and qualitative comparison with NTP is unclear. For example, the computational overhead for the weighting scheme and potential side effects (e.g., hallucinations) of Next-ToBE are not discussed.
- Experiments are confined to relatively small models (8B), and how well the proposed method scales to larger models is unclear. Evaluation of larger models could strengthen the claims.

**Questions:**

See weaknesses

---

> ### Author Response · Authors · 2025-11-21
> **Response to Reviewer 9ScA**
>
> We sincerely appreciate the reviewer's valuable and constructive comments, which have greatly helped us deepen our understanding and improve the quality of our paper. We have carefully considered and addressed each of your comments below.
>
> **Response to W1 - "Should include more baselines like ProphetNet, token order prediction, and label smoothing"**
>
> We thank the reviewer for this important comment. Following your suggestion, we further compared our method with additional baselines across five math reasoning benchmarks, including token order prediction (Zuhri et al., 2025) and label smoothing with a smoothing factor of 0.1 (Szegedy et al., 2016). For ProphetNet (Qi et al., 2020), which uses an encoder-decoder architecture while our model and the other MTP baselines are decoder-only, we retained the MTP-based Medusa as an architecturally aligned representative to avoid confounding effects on model performance.
>
> | Method | Math | Minerva_Math | OlympiadBench | AIME24 | AMC23 | AVG.   |
> |----------------------|:------:|:---------------:|:----------------:|:--------:|:--------:|:--------:|
> | Medusa (MTP)  | 66.0 | 20.6 | 29.0 | 10.0 | 47.5 | 34.6  |
> | Label Smoothing (NTP)    | 65.6 | 22.1 | 29.6 | 6.7  | 47.5 | 34.3  |
> | Token Order Prediction  | 67.8 | 22.8 | 31.3 | 10.0 | 50.0 | 36.4  |
> | Next-ToBE  | **68.2** |**23.2** |**31.9** |**13.3** | **55.0** | **38.3**  |
>
> As shown, Next-ToBE consistently outperforms all three baseline methods. The performance gains over label smoothing and token order prediction reach +4.0% and +1.9%, respectively. These results further underscore the advantages of Next-ToBE over a broader set of competitive baselines for complex reasoning tasks.
>
> **Responseto W2 - "Quantitative and qualitative comparison with NTP like computational overhead, side effect"**
>
> We thank the reviewer for this insightful comment, and our replies are listed below:
>
> * **Computational overhead**: Next-ToBE only incurs a modest overhead relative to the NTP baseline. For example,  on Qwen2.5-Math-1.5B (Section 4.4), Next-ToBE took **26 more minutes (+20%) and 3GB more memory (+32%)** against NTP; in comparison, among other MTP methods, Medusa took **41% more time and 75% more memory** against NTP, while **Mutor took 28% more time and 320% more memory (+30GB)** against NTP. Namely, our method achieved higher performance gains with lower computational overhead.
>
> * **Halucination**: By encouraging the model to fit a reasonably longer span rather than the immediate next token, Next-ToBE enhances global coherence and mitigates semantic drift. This is supported by empirical improvements in both token-level generative accuracy (Fig. 3b) and reasoning accuracy (Table 1 and Table 2), indicating a reduced tendency toward hallucination. For very long-horizon planning, however, the relationship between inference errors and hallucination can be more nuanced. We plan to investigate this further through in-depth analyses of chain-of-thought derivations, potentially leveraging strong process-reward models (e.g., Qwen2.5-Math-PRM-72B).
>
>
> **Response to W3 - "Evaluation of larger models"**
>
> We truly appreciate the reviewer for this important comment. To address this concern, we extended our study **from 8B models to a larger 14B-parameter model Qwen3-14B.** We then fine-tuned this base model using our method (Next-ToBE) as well as three baselines (NTP, MTP, and MuToR) and report their results on the five math reasoning benchmarks (average accuracy), consistent with the paper.
>
> | Method  | Math | Minerva_Math | OlympiadBench | AIME24 | AMC23 | AVG.|
> |-------------|:------:|:---------------:|:----------------:|:--------:|:--------:|:-----------:|
> | NTP    | 64.2 | 29.4 | 24.0| 3.3 | 40.0   | 32.2     |
> | MTP    | 64.6 | 30.1| 24.3| 3.3 | 35.0   | 31.5     |
> | MuToR  | 64.0 | 30.1 | 23.4 | **6.7** | 42.5   | 33.3     |
> | Next-ToBE| **65.4** | **30.5** | **24.6** | **6.7** |**47.5**  | **34.9**     |
>
> As observed, Next-ToBE continues to achieve the highest performance across all benchmarks on the 14B model. In terms of absolute gains, it outperforms MTP by 3.4%, NTP by 2.7%, and MuToR by 1.6%. These improvements are comparable or even larger than those observed on the smaller 8B base model. These results demonstrate that Next-ToBE exhibits strong scalability and maintains its effectiveness on larger models.

---

### Official Review · Reviewer_rAwF · 2025-11-04

**Soundness:** 3
**Presentation:** 3
**Contribution:** 3
**Rating:** 8
**Confidence:** 4

**Summary:**

This work proposes a novel training objective that adds a more learning signal at every time step by training to model to predict future tokens as well as the next token using the same output projection from the model. Empirically authors show how existing models are already capturing this sort of information about the future in the next token distribution and call it anitcipatory knowledge. Then they hypothesize that improving anticipatory capability in the next token distribution will improve generative performance of the model.

Authors design the objective with careful balancing between next token and future tokens learning; where they introduce weights that are based on the look-ahead distance as well as other scaling params.
Experiments show that their proposed method gives slightly better performance across the deck in most of benchmarks they consider compared to both next token and multi token prediction alternatives.

**Strengths:**

* well grounded motivation with empirical data point showing anticipatory knowledge already existing in the models.
* experiments are clearly described and adequate baselines and related work methods were considered.

**Weaknesses:**

* the balancing weights and added renormalization make the whole objective a bit cumbersome. Not very clear what might be the optimal value of K i.e. either we want to get as much anticipatory as possible or how do we optimize it?

* its known that MTP shows the substantial gains at scaling; so this method might as well show much more gains at scale. While this might be infeasible for authors due to compute constraints, absence of larger models and on more data is a minor weakness.

**Questions:**

How do you think anticipatory information can be further utilized ? e.g. during inference?

---

> ### Author Response · Authors · 2025-11-21
> **Response to Reviewer rAwF**
>
> We are sincerely grateful for the reviewer's valuable and inspiring comments, and for your insightful perspective, which has greatly guided us in improving the quality of our work. Below, we have carefully addressed each of your comments.
>
> **Response to Weakness - "Renormalization, weighting factor, and optimal window size $k$"**
>
> We thank the reviewer for these insightful comments, which have prompted us to reflect more deeply on these matters.
> 1. The **added re-normalization** ensures that the auxiliary loss remains numerically well-conditioned, so that the two loss terms in the final objective (Eq. (14)) are on comparable scales, making the balancing weight λ easier to tune. In experiments, we used a small $\lambda$ (~0.05). This allows the auxiliary loss to provide a reasonable level of longer-span fitting while simultaneously guaranteeing the dominance of the immediate next-token objective, which is benefitial to stable training.
> 2. The **window size $k$** is an important parameter. Fortunately, in our experiments across three tasks (code, math, QA) and two models (Qwen and LLaMA), the optimal $k$ lied in a relatively small range [8, 10] based on validation performances, which was easy to choose in practical applications. In the future, we plan to examine the impact of $k$ across more diverse tasks and larger models to obtain a more comprehensive understanding.
>
> In our experiments, we observe that moderate anticipatory modeling yields clear benefits, whereas an excessively large look-ahead window size $k$ dilutes the supervision signal and leads to degraded performance. This suggests a practical “**sweet spot**”, rather than a monotonic “**more anticipation is always better**” relationship. In the future, we will explore this relationship more thoroughly across different downstream tasks. We sincerely appreciate your insightful and inspiring comment on this point.
>
>
>
> **Response to W2 - "Scaling behaviour of the method on larger models"**
>
> We thank the reviewer for this important comment. To address your concern, we extended our study **from 8B models to a larger 14B-parameter model Qwen3-14B.** We then fine-tuned this base model using our method (Next-ToBE) as well as three baselines (NTP, MTP, and MuToR) and report their averaged accuracy results on the five math reasoning benchmarks (average accuracy), consistent with the original paper.
>
> | Method  | Math | Minerva_Math | OlympiadBench | AIME24 | AMC23 | AVG|
> |-------------|:------:|:---------------:|:----------------:|:--------:|:--------:|:-----------:|
> | NTP    | 64.2 | 29.4 | 24.0| 3.3 | 40.0   | 32.2     |
> | MTP    | 64.6 | 30.1| 24.3| 3.3 | 35.0   | 31.5     |
> | MuToR  | 64.0 | 30.1 | 23.4 | **6.7** | 42.5   | 33.3     |
> | Next-ToBE| **65.4** | **30.5** |**24.6**|**6.7** |**47.5** |**34.9**    |
>
> On this larger base model, Next-ToBE still achieves the **highest average accuracy**, with absolute gains of +2.7% over NTP and +3.4% over MTP - exceeding the improvements observed on the smaller 8B model. This demonstrates the scalability and potential of Next-ToBE. Furthermore, note that unlike MTP that relies on multiple auxiliary heads, Next-ToBE requires no architectural modification and thus scales smoothly with model size. We plan to conduct more extensive studies with larger models and broader datasets in future work.
>
> **Response to Question - "Can anticipatory information be used during inference"**
>
> We highly appreciate the reviewer for this inspiring angle. We believe anticipatory information can benefit inference, and we outline two preliminary thoughts below:
>
> 1.  **Integration with search algorithms**. The calibrated anticipatory signals generated by Next-ToBE can be integrated into search-based decoding methods, such as beam search or Monte Carlo Tree Search, to guide multiple reasoning trajectories. Each partial path can be scored using a combination of its cumulative log-probability and the confidence of predicted future token-bags, allowing the decoder to prune locally likely but globally inconsistent paths and preferentially expand coherent ones. This anticipatory-guided search can hopefully produce diverse, high-quality token sequences, improving answers in open-ended QA or generating candidate programs with higher execution success rates.
> 2.  **Ensemble anticipatory voting**. Additionally, the anticipatory signals of Next-ToBE can be exploited by aggregating predicted future distributions over several preceding steps - for example, through majority voting or consistency checks. This rich and collective decision may prioritize tokens better aligned with the model’s internal planning, thereby stabilizing next-token predictions, and mitigating premature divergence in chain-of-thought generation. maintain both local accuracy and global coherence in long-horizon generative tasks.

---

### Meta-Review · Area_Chair_AmVM · 2026-01-07

**Summary:**

Reviewers have raised the following main points.
- Pretraining applicability: Whether Next-ToBE is effective beyond post-training, i.e., when training models from scratch or with full-model pretraining rather than LoRA-style fine-tuning.
- Autoregressive stability: Whether encouraging future-token mass distorts the token-by-token generation distribution or causes drift during long autoregressive sampling.
- Computational efficiency and overhead: The true memory/runtime cost relative to NTP and strong MTP baselines, including the cost of the random-walk weighting.
- Scalability and hyperparameters: Sensitivity to multiple hyperparameters, perceived heuristic complexity, and whether gains persist at larger model scales and across stronger baselines.

Overall, the paper introduces a novel, well-motivated objective that consistently improves reasoning across tasks with favorable efficiency trade-offs. Major concerns were addressed well with new experiments and analyses.

**Reviewer Concerns:**

I do not believe any significant concerns are still outstanding.

**Reviewer Scores:**

I do not believe that the reviewers may have raised the scores further, as the scores already suffice for an acceptance (the last reviewer seems to have raised the score).

---

### Decision · Program_Chairs · 2026-01-26

Accept (Poster)